# Analysis of compositions of microbiomes with bias correction

Huang Lin ◉ [1] & Shyamal Das Peddada[1✉]

Differential abundance (DA) analysis of microbiome data continues to be a challenging problem due to the complexity of the data. In this article we define the notion of "sampling fraction" and demonstrate a major hurdle in performing DA analysis of microbiome data is the bias introduced by differences in the sampling fractions across samples. We introduce a methodology called Analysis of Compositions of Microbiomes with Bias Correction (ANCOM-BC), which estimates the unknown sampling fractions and corrects the bias induced by their differences among samples. The absolute abundance data are modeled using a linear regression framework. This formulation makes a fundamental advancement in the field because, unlike the existing methods, it (a) provides statistically valid test with appropriate p-values, (b) provides confidence intervals for differential abundance of each taxon, (c) controls the False Discovery Rate (FDR), (d) maintains adequate power, and (e) is computationally simple to implement.

[1] Department of Biostatistics, University of Pittsburgh, Pittsburgh, PA 15261, USA. ✉email: shyamal.peddada@nih.gov

A number of procedures have been proposed and used in the literature for identifying deferentially abundant taxa between two or more ecosystems. A detailed survey of some of the existing methods and their performance has been discussed in Weiss et al.[1]. As noted in a list of studies[2–6], the observed microbiome data are relative abundances which sum to a constant, hence they are compositional. Standard statistical methods are not appropriate for analyzing compositional data[7]. Methods such as ANOVA, Kruskal–Wallis test do not appropriately take into consideration the compositional feature of microbiome data when performing differential abundance (DA) analysis. As demonstrated in literatures[1,2], these methods are subject to inflated false discovery rates (FDR). Although meta-genomeSeq[8] was specifically developed for microbiome data, it too is subject to inflated FDR under the Gaussian mixture model[1,2].

ANCOM[2], which is based on Aitchison's methodology, uses relative abundances to infer about absolute abundances. According to an extensive simulation study[1], among the available methods for DA analysis, only ANCOM performs well in controlling FDR while maintaining high power, as long as the sample size is not too small. One of the deficiencies of ANCOM is that it does not provide $p$ value for individual taxon, nor can it provide standard errors or confidence intervals of DA for each taxon, and it can be computationally intensive.

The Differential Ranking (DR) methodology[6] reformulates the DA analysis as a multinomial regression problem. By imposing the Additive Log-Ratio transformation to relative abundances, the DR methodology accounts for compositionality of microbiome data. As demonstrated in[6], the ranks of relative differentials perfectly correlate with ranks of absolute differentials. However, similar to ANCOM, the DR procedure does not provide $p$ values or confidence intervals to declare statistical significance.

It is important to distinguish between absolute and relative abundances of taxa in a unit volume of an ecosystem. Change in the absolute abundance of a single taxon can alter the relative abundances of all taxa (Fig. 1). The choice of parameter for statistical analysis is important and needs to be clearly stated. Often researchers are interested in identifying taxa that are different in mean absolute abundance between two or more ecosystems[6]. Testing hypotheses regarding mean relative abundance is not equivalent to testing hypotheses regarding mean absolute abundance[2,6]. In addition, note that not all samples have the same sampling fraction, which is defined as the ratio of the expected absolute abundance of a taxon in a random sample (e.g., a stool sample) to its absolute abundance in a unit volume of the ecosystem (e.g., a unit volume of gut) where the sample was derived from. Consequently, the observed counts are not comparable between samples. Thus, all DA methodologies require the counts to be properly normalized to account for differences in sampling fractions across samples. Sampling fraction is affected by two components, namely, the microbial load in a unit volume of the ecosystem and the library size of the corresponding sample (e.g., total species abundances sequenced from a subject's stool sample). Therefore, it is not sufficient to normalize the library size across samples as one needs to take into consideration the differences in the microbial loads. Consider the toy example in Fig. 2. Suppose the gut of subject A as well as B consist of only two taxa, the red and green varieties. Clearly, the true absolute abundance of each taxon is 50% more in subject B's ecosystem as compared with subject A's. However, they each have the same library size (six each) in their respective samples. Furthermore, sample relative abundance as well as sample absolute abundances are identical in the two samples. If a normalization method is based only on the library size and ignores the sampling fraction, then the two samples would be considered as normalized. Consequently, an investigator would falsely conclude that none of the taxa are differentially abundant in the two ecosystems. This erroneous conclusion would be avoided if one recognizes that we have a larger sampling fraction in the sample obtained from A's

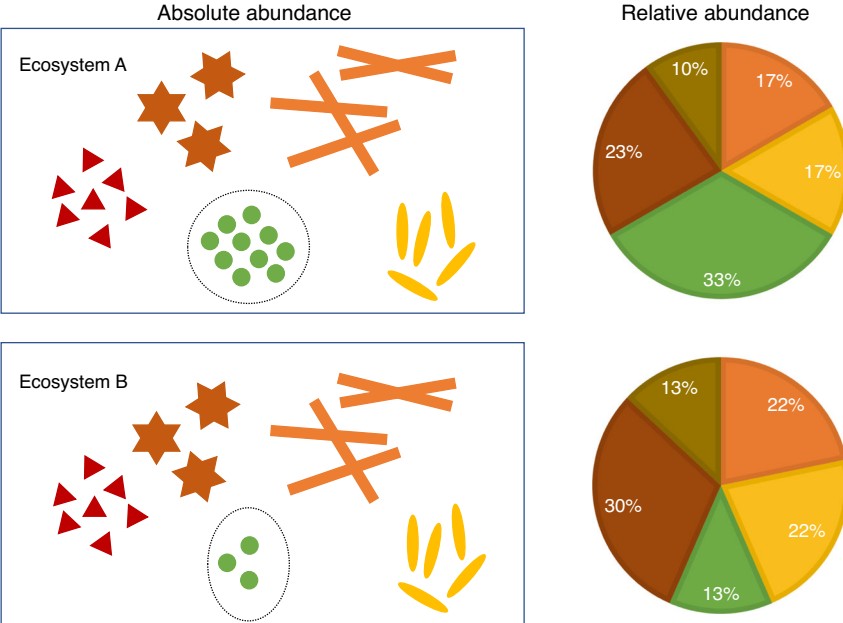

**Fig. 1 The distinction between absolute abundances and relative abundances.** As shown in this figure, all taxa (in different colors and shapes) may be identically abundant in a unit volume of two ecosystems (e.g., a unit volume of gut), except for one differentially abundant taxon (the green variety). Due to this one differentially abundant taxon, the two ecosystems may differ in the relative abundance of all taxa. A researcher may not only be interested in knowing if the mean relative abundance of a taxon is different between two ecosystems but may also want to know if the absolute abundance of a taxon is different in a unit volume of two ecosystems.

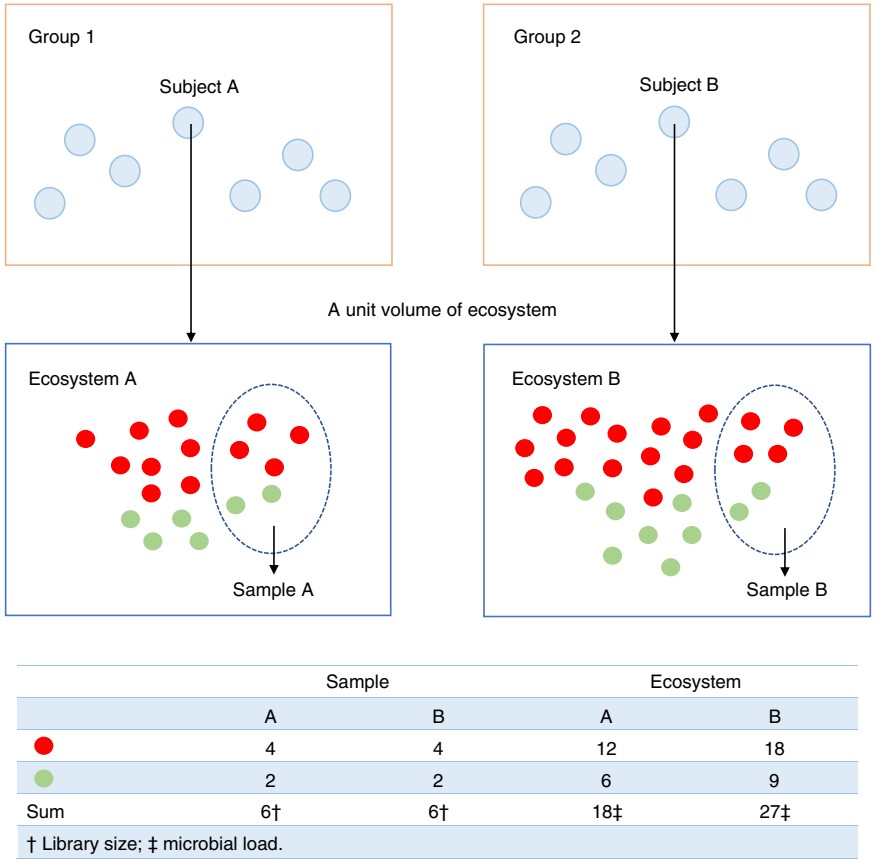

| | Sample | | Ecosystem | |
|---|---|---|---|---|
| | A | B | A | B |
| 🔴 | 4 | 4 | 12 | 18 |
| 🟢 | 2 | 2 | 6 | 9 |
| Sum | 6† | 6† | 18‡ | 27‡ |
| † Library size; ‡ microbial load. | | | | |

**Fig. 2 The bias introduced by cross-sample variations in sampling fractions.** Sampling fraction is defined as the ratio of expected absolute abundance in a sample to the corresponding absolute abundance in the ecosystem, which could be empirically estimated by the ratio of library size to the microbial load. Differences in sampling fractions may introduce bias and increase in false positive as well as false negative rates in differential abundance analysis. In this toy example, the microbial load for subject A in a unit volume of ecosystem (e.g., a unit volume of gut) is 18 (12 red + 6 green), while for subject B is 27 (18 red + 9 green). However, the samples taken from subject A and B have the same library size 6 (4 red + 2 green), the same observed absolute abundance as well as the same relative abundance of red and green taxa. Thus, one may mistakenly conclude that the red and green taxa are not differentially abundant, which is not the case in the two ecosystems. This false negative conclusion is caused by differences in the sampling fractions in the two samples. The sampling fraction in sample A is 3/9 and for B it is 2/9. One can similarly construct examples where a false positive conclusion is arrived at. Thus, a normalization method must account for differences in sampling fractions to avoid such erroneous conclusions.

ecosystem than from B's ($\frac{3}{9}$ vs $\frac{2}{9}$), Thus, normalizing data on the basis of sampling fractions gives a better description of the truth than normalization methods that rely purely on the library sizes.

Ideally, under the null hypothesis, the test statistic for DA analysis should be (at least approximately) centered at zero (i.e., unbiased). However, for many DA methods, this is not always true for at least one of the following reasons: (1) The test statistic may not be designed for testing hypothesis regarding the actual parameter of interest; (2) Data are not properly normalized; (3) Underlying structure, such as compositionality, is ignored. Motivated by the limitations of existing DA methods, in this paper we propose a methodology called Analysis of Compositions of Microbiomes with Bias Correction (ANCOM-BC) that is aimed to address the problems mentioned above. As in ANCOM and DR, the proposed ANCOM-BC methodology assumes that the observed sample is an unknown fraction of a unit volume of the ecosystem, and the sampling fraction varies from sample to sample. ANCOM-BC accounts for sampling fraction by introducing a sample-specific offset term in a linear regression framework, that is estimated from the observed data. The offset term serves as the bias correction, and the linear regression framework in log scale is analogous to log-ratio transformation to deal with the compositionality of microbiome data. The case of zero counts

is also discussed in "Methods" section. This methodology has some conceptual similarities with DR, but is fundamentally different. With ANCOM-BC, one can perform standard statistical tests and construct confidence intervals for DA. Moreover, as demonstrated in benchmark simulation studies, ANCOM-BC (a) controls the FDR very well while maintaining adequate power compared with other popular methods, and (b) it is substantially faster than ANCOM. The CPU time (RStudio, x86_64-apple-darwin15.6.0, and macOS) is 0.28 min vs. 63 min when the number of taxa is 500. The CPU time for ANCOM increases dramatically as the number of taxa increases to 1000. In this case, the CPU times for ANCOM-BC and ANCOM are 0.51 and 211 min, respectively. In addition to results based on synthetic data, we also illustrate ANCOM-BC using the well-known global gut microbiota dataset[9].

## Results

**Normalization.** Using simulated data, we illustrate how the existing normalization methods fail to eliminate the bias introduced by differences in sampling fractions across samples, whereas the normalization method introduced in ANCOM-BC performs well. Specifically, we compare our proposed method

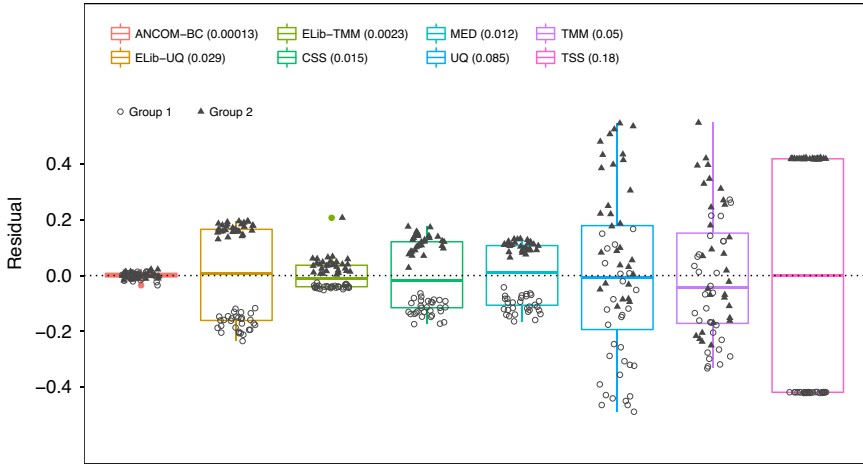

**Fig. 3 Box plot of residuals between true sampling fraction and its estimate for each sample.** In the box plot, the lower and upper hinges correspond to the first and third quartiles (the 25th and 75th percentiles). The median is represented by a solid line within the box. The upper whisker extends from the hinge to the largest value (maxima) no further than 1.5 times Interquartile Range (IQR, distance between the first and third quartiles) from the hinge, the lower whisker extends from the hinge to the smallest value (minima) at most 1.5 times IQR of the hinge. Data beyond the end of the whiskers are called "outlying" points. $N = 30$ samples examined over two experimental groups (denoted by circles and triangles) and the data points are overlaid in each box. Text on the upper left corner indicates the color for each method and variances are provided within parenthesis for each method. The variability in sampling fractions is set to be large. An ideal box plot should display a narrow height (i.e., smaller variability) and samples from the two groups should be intermixed and not display any systematic separation. We note that all existing methods have larger variances compared with ANCOM-BC, and TSS has the largest variance. Except ANCOM-BC, UQ, and TMM, we see from the plot that circles and triangles are systematically separated, which indicates that ELib-UQ, ELib-TMM, CSS, MED, and TSS do not account for systematic bias due to differences in sampling fractions across groups.

with Cumulative-Sum Scaling (CSS) implemented in metagenomeSeq[8], Median (MED) in DESeq2[10], Upper Quartile (UQ) and Trimmed Mean of $M$ values (TMM), and Total-Sum Scaling (TSS). In addition, we also considered modified versions of UQ and TMM implemented in edgeR[11]. These are obtained by multiplying the normalization factors with the corresponding library size to account for "effective library size"[12], and are denoted as ELib-UQ and ELib-TMM (see Supplementary Table 7 for formulas and Supplementary Fig. 11 for workflow).

We considered a variety of simulation scenarios as follows. The details of the simulation study are presented in the Supplementary Notes.

(1) Unbalanced microbial load in two experimental groups and balanced library size for each sample. This results in a large variability in sampling fractions (Fig. 3).

(2) Unbalanced microbial load in two experimental groups and unbalanced library size for each sample. This results in a moderate variability in sampling fractions (Supplementary Fig. 1).

(3) Balanced microbial load in two experimental groups and balanced library size for each sample. This results in a small variability in sampling fractions (Supplementary Fig. 2).

Thus, we simulated data where sampling fraction in Group 1 is systematically different from sampling fraction in Group 2. Consequently, observed absolute abundances in the samples in the two groups were systematically different even though the actual absolute abundances in the ecosystems are same. To evaluate the performance of each normalization method, we introduced a residual measure that estimates the deviation between the estimated sampling fraction and the true sampling fraction (see Supplementary Discussion). For simplicity of exposition, we plotted the centered residuals, by subtracting the group average of the residuals. If a normalization method is effective then it should eliminate the bias due to the differences in the sampling fractions so that samples from the two groups

(circles and triangles) in Fig. 3 should intermix and not cluster by the group labels.

From Fig. 3 (and Supplementary Figs. 1, 2) we notice that the samples normalized by ANCOM-BC are nicely intermixed and do not cluster by the group labels. This is not the case with most of the remaining methods where residuals cluster by group labels, thus indicating that they are unable to eliminate the underlying differences in sampling fractions between the two groups. Thus, under the null hypothesis of no difference in the absolute abundance of a taxon in two groups, their test statistics are not centered at zero. This results in inflated FDR (see Supplementary Discussion). We also note from Fig. 3 and Supplementary Figs. 1, 2, that not only ANCOM-BC does well in estimating the bias due to differences in sampling fraction, the variability in the estimates of the sampling fractions is very small as seen from the height of the box plot for ANCOM-BC. This is an important observation because it suggests that the variability in the estimator of bias due to sampling fraction is potentially negligible in the test statistic described in "Methods" section.

Clearly, as seen in Fig. 4a, b and Supplementary Figs. 3a, b, 4a, b, the normalization of data has a major effect on the FDR and power of various methods.

**DA analyses.** Simulating data from Poisson-Gamma distributions (see Supplementary Notes for simulation settings and Supplementary Fig. 12 for workflow), we evaluated the performance of various methods in terms of FDR and power. Since there is no hard threshold available for DR to declare whether a taxon is differentially abundant or not, it was not included in this simulation study.

Not surprisingly, standard Wilcoxon rank-sum test applied to relative abundance data leads to highly inflated FDR (Fig. 4a and Supplementary Figs. 3a, 4a) in all simulation scenarios. This is primarily because such standard tests ignore the compositional structure of the data, and seen from Fig. 3, TSS does not successfully normalize the data. Simply applying nonparametric

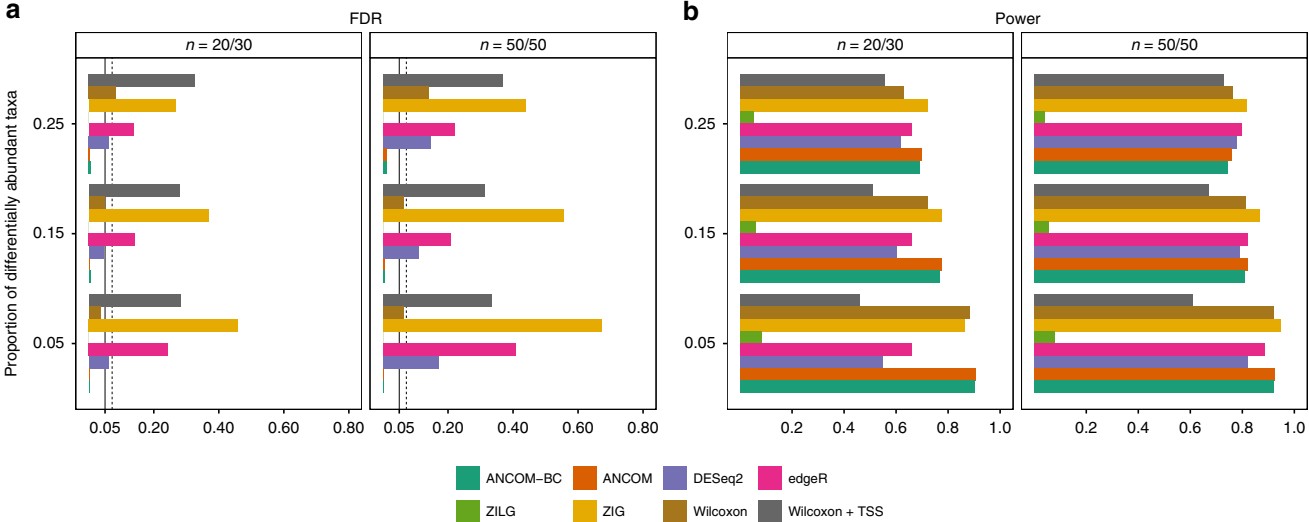

**Fig. 4 FDR and power comparisons using synthetic data from Poisson-Gamma distributions.** The False Discovery Rate (FDR) and power of various differential abundance (DA) analyses (two-sided) are shown in **a** and **b**, respectively. The variability in sampling fractions is set to be large. The Y-axis denotes patterns of proportion of differentially abundant taxa. The solid vertical line is the 5% nominal level of FDR, and the dashed vertical line denotes 5% nominal level plus one standard error (SE). By default, ANCOM-BC implements Bonferroni correction and other DA methods implement BH procedure to adjust for multiple comparisons. Color and the name of the corresponding DA method are shown at the bottom within the graph. Two simulation scenarios are considered: small and unbalanced data ($n_1 = 20$, $n_2 = 30$), as well as large and balanced data ($n_1 = n_2 = 50$); number of simulations = 100. Results show that only ANCOM and ANCOM-BC control the FDR under the nominal level (5%) while maintaining power comparable with other methods. Gaussian model version of metagenomeSeq has highly inflated FDR, while the log-Gaussian version has substantial loss of power, sometimes well below 5%. Other than ANCOM-BC and ANCOM, as the sample size within each group increases, so does the FDR for all other existing methods.

tests without any normalization can also be problematic when the sampling fractions are different across experimental groups (Fig. 4a). The two widely used count-based methods in RNA-Seq literature, edgeR (implemented using ELib-TMM[12] by default) and DESeq2, generally exceed the 5% nominal FDR level when there are differences in sampling fractions (Fig. 4a and Supplementary Fig. 3a). For instance, edgeR has FDR as large as 40% (Fig. 4a), meaning that 40% of findings could be potentially false discoveries. The zero-inflated Gaussian mixture model used in metagenomeSeq (ZIG) consistently has the largest FDR when sampling fractions are not constant (Fig. 4a and Supplementary Fig. 3a). In some cases, the FDR could be as much as 70%, which perhaps is partly due to the Gaussian distribution assumption for log abundance data. Although metagenomeSeq using zero-inflated Log-Gaussian mixture model successfully controls the FDR under 5% in all simulations, it suffers a severe loss of power (Fig. 4b and Supplementary Figs. 3b, 4b). The power of detecting differentially abundant taxa could be lower than 10%.

Similar to ANCOM, ANCOM-BC not only controls the FDR at the nominal level (5%) but also maintains adequate power in all simulation settings considered here. An important observation to be made regarding all methods, other than ANCOM and ANCOM-BC, is that as the sample size within each group increases, so does the FDR. This is perhaps a consequence of the fact that the test statistics are not centered at the true null parameter but are shifted due to differences in the sampling fraction. Hence asymptotically, these tests fail to control the false positive as well as FDR (see Supplementary Discussion).

In addition to the above Poisson-Gamma model, we performed simulations using the real global patterns data[13], to get a broader perspective on the performance of the various methods (see Supplementary Notes for simulation details). In this case again, ANCOM and ANCOM-BC controlled the FDR and competed well in terms of power with all other methods. The estimated FDR of DESeq2 and edgeR increased further in this simulation set-up (Supplementary Fig. 5a, b) compared with the simulation using Poisson-Gamma distribution. Note that DESeq2 and edgeR were designed for Poisson-Gamma distribution, and hence it is not surprising that these methods performed poorly in this new set-up.

**Illustration using gut microbiota data.** We illustrate ANCOM-BC by analyzing the US, Malawi and Venezuela gut microbiota data[9]. This dataset consists of 11,905 OTUs obtained from subjects in the USA ($n = 317$), Malawi ($n = 114$), and Venezuela ($n = 99$). We first assessed the performance of different normalization methods mentioned above. One heuristic approach to gain insights on the impact of normalization is to examine how well the normalized samples separate from each other according to their phenotypes in a nonmetric multidimensional scaling (NMDS) plot. We provide the results for Malawi and Venezuela populations in Fig. 5.

As seen from this figure, ANCOM-BC appears to perform very well visually in separating samples from the two populations and has the largest between-group sum of squares (BSS). BSS measures how well clusters are separated. Larger the BSS value the better a method is in clustering objects according to group labels. ELib-TMM, CSS, and MED also performed well. Consistent with the bias correction and FDR/Power simulations reported in Figs. 3 and 4, where ELib-UQ, UQ, TMM, and TSS perform poorly in correcting biases and have poor FDR control, they also have poor performances in distinguishing samples based on their nationalities.

We also report results of pairwise DA analyses at phylum level among the above three countries using ANCOM-BC. It is well-known that the infant gut microbiota evolve with their age[14] due to changes in the feeding patterns, diet, and other exposures. Hence, for illustration purposes, we performed a stratified analysis by considering two age groups, infants below 2 years (labeled as "infants") and adults between 18 and 40 (labeled as

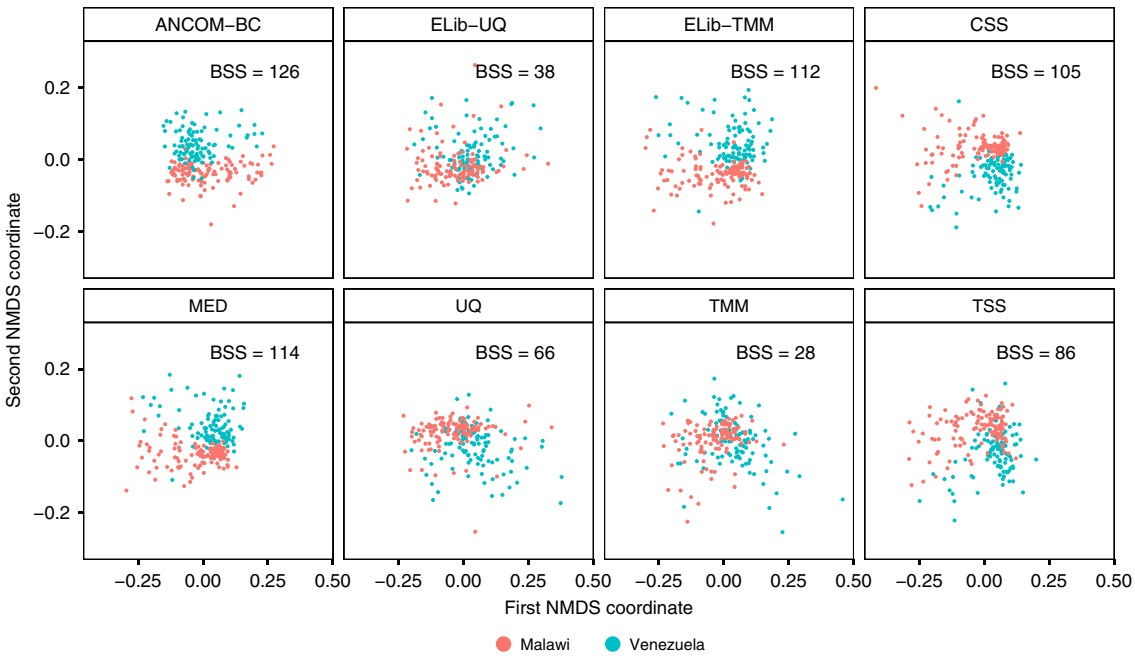

**Fig. 5 Non-metric multidimensional scaling (NMDS) visualizations of normalized data.** First two NMDS coordinates are used to evaluate the performance of various normalization methods (ANCOM-BC, ELib-UQ, ELib-TMM, CSS, MED, UQ, TMM, and TSS) applied on Malawi and Venezuela samples of the global gut microbiota data at genus level. Samples from Malawi are colored in red while samples from Venezuela are colored in green. Visually, ANCOM-BC, ELib-TMM, MED, and CSS appear to provide best separation between Malawi and Venezuela samples. Quantitatively, in terms of Between-Group Sum of Squares (BSS), standardized by Total Sum of Squares (TSS) so that all methods are comparable, ANCOM-BC has the largest BSS, followed by ELib-TMM, and MED. Rest of the methods perform poorly both visually as well as quantitatively with small BSS values. These findings appear to be consistent with results of the synthetic data shown in Fig. 3 and Supplementary Figs. 1, 2.

"adults"). Results of all pairwise comparisons are provided in Fig. 6a and Supplementary Table 1. Note that ANCOM-BC is the first method in the literature that can not only identify differentially abundant taxa while controlling the FDR for multiple testing, it also provides 95% simultaneous confidence intervals for the mean DA of each taxon in the two experimental groups. These confidence intervals are adjusted for multiplicity using Bonferroni method. Thus, a researcher can evaluate the effect size associated with each taxon when comparing two experimental groups. This is particularly important in the present climate when researchers are increasingly skeptical about making decisions based on $p$ values (alone)[15].

Interestingly, phyla such as Cyanobacteria, Elusimicrobia, Euryarchaeota, and Spirochetes, which are known to be associated with rural environment and hygiene[16–19], are significantly more abundant among Malawi than the US infants and adults. We discover an interesting trend in the absolute abundance of phylum Verrucomicrobia, whose absolute abundance is known to increase with antibiotics usage to protect against pathogens and other opportunistic bacteria[20]. Consistent with the high usage of antibiotics in the western world among infants as well as adults, we discover a significant increase in the absolute abundance of Verrucomicrobia in US relative to Malawi adults and infants, and relative to Venezuelan adults (Fig. 6a). Similarly, there is a significant increase in its absolute abundance among Venezuelan infants compared with Malawi (Fig. 6a).

It is well-documented in the literature that BMI is linked to the ratio of Bacteroidetes to Firmicutes[21]. In our sample, the US infants, as well as adults, had higher BMI than their counterparts in Malawi; The US infants also had higher BMI than Venezuela infants (Supplementary Table 2). Interestingly the ratio of Bacteroidetes to Firmicutes was larger among Malawi infants

than the US infants (Fig. 6b and Supplementary Table 3). Similarly, the ratio was significantly larger among Venezuela infants than the US infants (Fig. 6b and Supplementary Table 3). Although the differences of the ratio of Bacteroidetes to Firmicutes between US and non-US adults were not significant, the effect sizes showed a similar trend as infants indicating that US adults had smaller ratio of Bacteroidetes to Firmicutes. We did not find any significant differences between Malawi and Venezuelan infants as well as adults. These results are in line with our findings that there were no differences in the mean absolute abundances of Firmicutes as well as Bacteroidetes among Malawi and Venezuelan infants as well as adults (Fig. 6a).

## Discussion

The DA analysis of microbiome data is a challenging problem[5,6], in part due to inaccessibility of data necessary for drawing inferences on DA in two or more ecosystems. An important unobservable parameter that impacts DA analysis is the sampling fraction of a sample drawn from a unit volume of ecosystem. As noted in previous studies[5,6], the bias correction due to sampling fraction is a major hurdle. While, ANCOM as well as DR procedures find ways to get around the problem from different perspectives, there is room for improvement. Secondly, differential relative abundance analysis of microbiome data is not equivalent to differential absolute abundance analysis of microbiome data. Often simplex or compositional data analysis-based methods transform the simplex coordinate system to Euclidean space by performing log ratio transformation. However, such methods require the researcher to prespecify the reference taxon and the results may be highly dependent on the choice of the reference taxon[6]. It is important to reiterate that ANCOM computes log-ratios with respect to all taxa and thus the choice of

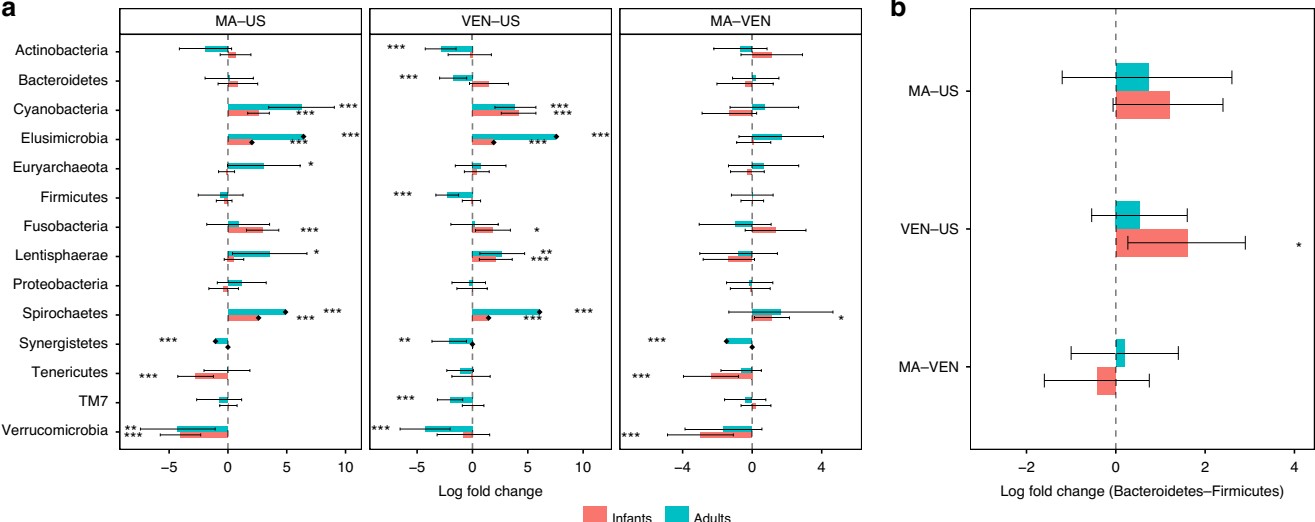

**Fig. 6 Analysis of the global gut microbiota data in phylum level.** Data are represented by effect size (log fold change) and 95% confidence interval bars (two-sided; Bonferroni adjusted) derived from the ANCOM-BC model. All effect sizes with adjusted $p < 0.05$ are indicated, *significant at 5% level of significance; **significant at 1% level of significance; ***significant at 0.1% level of significance. Exact adjusted $p$ values can be found in Supplementary Table 1. Diamonds on top of some bars indicate structural zeros. **a** Pairwise differential abundance analyses stratified by age using ANCOM-BC: Infants (age $\leq 2$ years, $n = 133$), and adults ($18 \leq$ age $\leq 40$, $n = 83$). Infant samples are colored in red and adult samples are colored in green. Phyla Acidobacteria and Chloroflexi are not represented in the plot since they are present only in Venezuela samples. **b** Pairwise tests using ANCOM-BC for the equality of mean log ratio of Bacteroidetes to Firmicutes stratified by age. Viewing **b** and Supplementary Table 2 together, lower BMI seems to be associated with higher levels of the Bacteroidetes to Firmicutes ratio, a result widely acknowledged in the literature.

reference taxon is not an issue for ANCOM. As demonstrated mathematically in ANCOM methodology[2], as long as two taxa are not differentially abundant between two ecosystems, one can draw inferences about DA using differential relative abundance.

ANCOM-BC enjoys several important unique characteristics. First, it is the only method available in the literature that estimates the sampling fraction and performs DA analysis by correcting bias due to differential sampling fractions across samples. It is the only procedure that provides valid $p$ values and confidence intervals for each taxon. Second, unlike ANCOM, it simplifies DA analysis by recasting the problem as a linear regression problem with an offset. The offset is due to the sampling fraction. By virtue of linear regression formulation, ANCOM-BC can be applied to a broad collection of study designs, including longitudinal data, repeated measurements design, covariance adjusted analysis, and so on. Using a broad range of simulations studies, we demonstrate that ANCOM-BC, like ANCOM, controls the FDR very well, while almost all other methods investigated in this paper fail.

The ANCOM-BC methodology may not perform well when the sample sizes are very small, such as $n = 5$ per group. The FDR is not controlled by ANCOM-BC in such cases (Supplementary Fig. 6a, b). However, when the sample size increases to 10, our simulation results indicate that ANCOM-BC controls FDR with adequate power (Supplementary Fig 6a, b). We also evaluated the performance of ANCOM-BC when the number of taxa is small, as when researchers perform DA analysis at the phylum or class levels. Even in such instances, ANCOM-BC controls the FDR very well while maintaining high power (Supplementary Table 4). ANCOM-BC performs best in terms of FDR control when the proportion of differentially abundant taxa is not too large (e.g., <75%). Otherwise, it may have slightly elevated FDR (Supplementary Fig. 7a, b). However, none of the other methods control the FDR either, in fact, they have larger FDRs than ANCOM-BC.

In many applications, researchers are interested in drawing inferences regarding DA of taxa in more than two ecosystems. We extended ANCOM-BC to deal with such multigroup situations. Extensive simulations suggest that ANCOM-BC controls FDR while maintaining high power (Supplementary Table 5).

In summary, the proposed ANCOM-BC methodology (1) explicitly tests hypothesis regarding differential absolute abundance of individual taxon and provides valid confidence intervals; (2) provides an approach to correct the bias induced by (unobservable) differential sampling fractions across samples; (3) takes into account the compositionality of the microbiome data, and (4) does not rely on strong parametric assumptions. With the linear regression framework adopted in ANCOM-BC, it allows researchers to derive $p$ value associated with each taxon as well as confidence interval estimation for differential absolute abundance. These are unique to ANCOM-BC, to the best of our knowledge. Last but not the least, because of the regression framework adopted in ANCOM-BC, it can be extended to more general settings involving multigroup comparisons, adjusting covariates as well as applying to longitudinal/repeated measurements data.

## Methods
**Notation.** The notations described in ANCOM-BC methodology are summarized in Table 1.

**Data preprocessing.** We adopted the methodology of ANCOM-II[22] as the preprocessing step to deal with different types of zeros before performing DA analysis.

There are instances where some taxa are systematically absent in an ecosystem. For example, there may be taxa present in a soil sample from a desert that might absent in a soil sample from a rain forest. In such cases, the observed zeros are called structural zeros. Let $p_{ij}$ denote proportion non-zero samples of the $i$th taxon in the $j$th group, and let $\hat{p}_{ij} = \frac{1}{n_j} \sum_{k=1}^{n_j} I(O_{ijk} \neq 0)$ denote the estimate of $p_{ij}$. In practice, we declare the $i$th taxon to have structural zeros in the $j$th group if either of the following is true:

**Table 1 Summary of notations.**

| Notation | Description |
|---|---|
| $i$ | Taxon index, $i = 1, 2, ..., m$. |
| $j$ | Group index, $j = 1, 2, ..., g$. |
| $k$ | Sample index, $k = 1, 2, ..., n_j$. |
| $\theta_{ij}{}^{a}$ | Expected absolute abundance of $i$th taxon in a unit volume of ecosystem in the $j$th group. |
| $A_{ijk}{}^{b}$ | Unobserved absolute abundance of $i$th taxon in a unit volume of ecosystem of $k$th sample in the $j$th group. |
| $A_{\cdot jk}{}^{b}$ | Microbial load in a unit volume of ecosystem of $k$th sample in the $j$th group. $A_{\cdot jk} = \sum_{i=1}^{m} A_{ijk}$. |
| $\gamma_{ijk}{}^{b}$ | Unobserved relative abundance of $i$th taxon in a unit volume of ecosystem of $k$th sample in the $j$th group. $\gamma_{ijk} = \frac{A_{ijk}}{A_{\cdot jk}}$. |
| $O_{ijk}{}^{b}$ | Observed absolute abundance of $i$th taxon in a random specimen taken from a unit volume of ecosystem of $k$th sample in the $j$th group. |
| $O_{\cdot jk}{}^{b}$ | Library size of a random specimen taken from a unit volume of ecosystem of $k$th sample in the $j$th group. $O_{\cdot jk} = \sum_{i=1}^{m} O_{ijk}$. |
| $r_{ijk}{}^{b}$ | Observed relative abundance of $i$th taxon in a random specimen taken from a unit volume of ecosystem of $k$th sample in the $j$th group. $r_{ijk} = \frac{O_{ijk}}{O_{\cdot jk}}$. |
| $c_{jk}{}^{a}$ | For $k$th sample from the $j$th group, $c_{jk}$ represents the proportion of its ecosystem (unobserved absolute abundance) in a random sample (observed absolute abundance), thus $c_{jk} = \frac{E(O_{ijk}\|A_{ijk})}{A_{ijk}}$. We shall refer to this constant as "sampling fraction". |
| $y_{ijk}{}^{b}$ | $\log(O_{ijk})$. |
| $\mu_{ij}{}^{a}$ | $\log(\theta_{ij})$. |
| $d_{jk}{}^{a}$ | $\log(c_{jk})$. |

$^{a}$Parameter.
$^{b}$Random variable.

(a)  $\hat{p}_{ij} = 0$.
(b)  $\hat{p}_{ij} - 1.96\sqrt{\frac{\hat{p}_{ij}(1-\hat{p}_{ij})}{n_j}} \leq 0$.

If a taxon is considered to be a structural zero in an experimental group then, for that specific ecosystem, the taxon is not used in further analysis. Thus, suppose there are three ecosystems A, B, and C and suppose *taxon X* is a structural zero in ecosystems A and B but not in C, then *taxon X* is declared to be differentially abundant in C relative to A and B and not analyzed further. If *taxon Y* is a structurally zero in ecosystem A but not in B and C, in that case we declare that *taxon Y* is differentially abundant in B relative to A as well as differentially abundant in C relative to A. We then compare the absolute abundance of *taxon Y* between B and C using the methodology described in this section. Taxa identified to be structural zeros among all experimental groups are ignored from the following analyses.

In a similar fashion, we address the outlier zeros as well as sampling zeros using the methodology developed in ANCOM-II[22].

**Model assumptions**. **Assumption 0.1.**

$$E(O_{ijk}|A_{ijk}) = c_{jk}A_{ijk}$$
$$Var(O_{ijk}|A_{ijk}) = \sigma^2_{w,ijk}, \tag{1}$$

where $\sigma^2_{w,ijk}$ = variability between specimens *within* the $k$th sample from the $j$th group. Therefore, $\sigma^2_{w,ijk}$ characterizes the within-sample variability. Typically, researchers do not obtain more than one specimen at a given time in most microbiome studies. Consequently, variability between specimens within sample is usually not estimated.

According to Assumption 0.1, in expectation the absolute abundance of a taxon in a random sample is in constant proportion to the absolute abundance in the ecosystem of the sample. In other words, the expected relative abundance of each taxon in a random sample is equal to the relative abundance of the taxon in the ecosystem of the sample.

**Assumption 0.2.** For each taxon $i$, $A_{ijk}$, $j = 1, ..., g$, $k = 1, ..., n_j$, are independently distributed with

$$E(A_{ijk}|\theta_{ij}) = \theta_{ij}$$
$$Var(A_{ijk}|\theta_{ij}) = \sigma^2_{b,ij}, \tag{2}$$

where $\sigma^2_{b,ij}$ = between-sample variation within group $j$ for the $i$th taxon.

The Assumption 0.2 states that for a given taxon, all subjects within and between groups are independent, where $\theta_{ij}$ is a fixed parameter rather than a random variable.

**Regression framework**. From Assumptions 0.1 and 0.2, we have:

$$E(O_{ijk}) = c_{jk}\theta_{ij}$$
$$Var(O_{ijk}) = f(\sigma^2_{w,ijk}, \sigma^2_{b,ij}) := \sigma^2_{t,ijk}. \tag{3}$$

Motivated by the above set-up, we introduce the following linear model framework for log-transformed OTU counts data:

$$y_{ijk} = d_{jk} + \mu_{ij} + \epsilon_{ijk}, \tag{4}$$

with

$$E(\epsilon_{ijk}) = 0,$$
$$E(y_{ijk}) = d_{jk} + \mu_{ij}, \tag{5}$$
$$Var(y_{ijk}) = Var(\epsilon_{ijk}) := \sigma^2_{ijk}.$$

Note that with a slight abuse of notation for simplicity of exposition, the above log-transformation of data is inspired by the Box–Cox family of transformations[23] which are routinely used in data analysis. Note that $d$ in the above equation is not exactly $log(c)$ due to Jensen's inequality, it simply reflects the effect of $c$

An important distinction from standard ANOVA: Before we describe the details of the proposed methodology, we like to draw attention to a fundamental difference between the current formulation of the problem and the standard one-way ANOVA model. For simplicity, let us suppose we have two groups, say a treatment and a control group. Let us also suppose that there is only one taxon in our microbiome study and $n$ subjects are assigned to the treatment group and $n$ are assigned to the control group. Suppose the researcher is interested in comparing the mean absolute abundance of the taxon in the ecosystems of the two groups. Then under the above assumptions, the model describing the study is given by:

$$y_{jk} = d_{jk} + \mu_j + \epsilon_{jk}, j = 1, 2, k = 1, 2, \dots, n.$$

Then trivially the sample mean absolute abundance of $j$th group is given by $\bar{y}_{j\cdot} = \frac{1}{n}\sum_{k=1}^{n} y_{jk}$ and $E(\bar{y}_{j\cdot}) = \frac{1}{n}\sum_{k=1}^{n} d_{jk} + \mu_j = \bar{d}_{j\cdot} + \mu_j$. The difference in the sample means between the two groups is $\bar{y}_{1\cdot} - \bar{y}_{2\cdot}$ and its expectation is $E(\bar{y}_{1\cdot} - \bar{y}_{2\cdot}) = (\bar{d}_{1\cdot} - \bar{d}_{2\cdot}) + (\mu_1 - \mu_2)$. Under the null hypothesis $\mu_1 = \mu_2$, $E(\bar{y}_{1\cdot} - \bar{y}_{2\cdot}) = \bar{d}_{1\cdot} - \bar{d}_{2\cdot} \neq 0$, unless $\bar{d}_{1\cdot} = \bar{d}_{2\cdot}$. Thus because of the differential sampling fractions, which are sample specific, the numerator of the standard $t$-test under the null hypothesis for these microbiome data is non-zero. This introduces bias and hence inflates the Type I error. On the other hand, the standard one-way ANOVA model for two groups, which is not applicable for the microbiome data described in this paper, is of the form:

$$y_{jk} = d + \mu_j + \epsilon_{jk}, j = 1, 2, k = 1, 2, \dots, n.$$

Hence under the null hypothesis $\mu_1 = \mu_2$, $E(\bar{y}_{1\cdot} - \bar{y}_{2\cdot}) = 0$. Thus, in this case the standard $t$-test is appropriate. Hence in this paper we develop methodology to eliminate the bias introduced by the differential sampling fraction by each sample. To do so, we exploit the fact that we have a large number of taxa on each subject and we borrow information across taxa to estimate this bias, which is the essence of the following methodology.

Bias and variance of bias estimation under the null hypothesis: From the above model (equation (4)), for each $j$, note that $E(\bar{y}_{ij\cdot}) = \bar{d}_{j\cdot} + \mu_{ij}$ and $E(\bar{y}_{\cdot jk}) = d_{jk} + \bar{\mu}_{\cdot j}$, where $\bar{w}$ represents the arithmetic mean over the suitable index. Using the least squares framework, we therefore estimate $\mu_{ij}$ and $d_{jk}$ as follows:

$$\hat{d}_{jk} = \bar{y}_{\cdot jk} - \bar{y}_{\cdot j\cdot}, k = 1, \dots, n_j, j = 1, 2, \dots g,$$
$$\hat{\mu}_{ij} = \bar{y}_{ij\cdot} - \bar{\bar{d}}_{j\cdot} = \bar{y}_{ij\cdot}, i = 1, \dots, m. \tag{6}$$

Note that $E(\hat{\mu}_{ij}) = E(\bar{y}_{ij\cdot}) = \mu_{ij} + \bar{d}_{j\cdot}$. Thus, for each $j = 1, 2, \dots g$, $\hat{\mu}_{ij}$ is a biased estimator and $E(\hat{\mu}_{i1} - \hat{\mu}_{i2}) = (\mu_{i1} - \mu_{i2}) + \bar{d}_{1\cdot} - \bar{d}_{2\cdot}$. Denote $\delta = \bar{d}_{1\cdot} - \bar{d}_{2\cdot}$. To begin with, in the following we shall assume there are two experimental groups

with balanced design, i.e., $g = 2$ and $n_1 = n_2 = n$. Later the methodology is easily extended to unbalanced design and multigroup settings. Suppose we have two ecosystems and for each taxon $i$, $i = 1, 2, \ldots m$, we wish to test the hypothesis

$$H_0 : \mu_{i1} = \mu_{i2}$$
$$H_1 : \mu_{i1} \neq \mu_{i2}. \tag{7}$$

Under the null hypothesis, $E(\hat{\mu}_{i1} - \hat{\mu}_{i2}) = \delta \neq 0$, and hence biased. The goal of ANCOM-BC is to estimate this bias and accordingly modify the estimator $\hat{\mu}_{i1} - \hat{\mu}_{i2}$ so that the resulting estimator is asymptotically centered at zero under the null hypothesis and hence the test statistic is asymptotically centered at zero. First, we make the following observations. Since $E(\overline{y}_{ij \cdot}) = \overline{d}_{j \cdot} + \mu_{ij}$ and $\hat{\mu}_{ij} = \overline{y}_{ij \cdot}$, therefore $\hat{\mu}_{ij}$ is an unbiased estimator of $\overline{d}_{j \cdot} + \mu_{ij}$. From (5) and Lyapunov central limit theorem, we have:

$$\frac{\hat{\mu}_{ij} - (\mu_{ij} + \overline{d}_{j \cdot})}{\sigma_i} \to_d N(0, 1) \ \text{as} \ n \to \infty, \tag{8}$$
$$\text{where} \ \sigma_{ij}^2 = Var(\hat{\mu}_{ij}) = Var(\overline{y}_{ij \cdot}) = \frac{1}{n^2} \sum_{k=1}^{n} \sigma_{ijk}^2.$$

Let $\Sigma_{jk}$ denote an $m \times m$ covariance matrix of $\epsilon_{jk} = (\epsilon_{1jk}, \epsilon_{2jk}, \ldots, \epsilon_{mjk})^T$, where $\sigma_{ii'jk}$ is the $(i, i')^{th}$ element of $\Sigma_{jk}$ and $\sigma_{ijk}^2$ is the $i$th diagonal element of $\Sigma_{jk}$. Furthermore, suppose

**Assumption 0.3.**

$$\sigma_{ijk}^2 < \sigma_0^2 < \infty$$
$$\frac{\sum_{i \neq i'}^{m} \sigma_{ii'jk}}{m^2} = o(1). \tag{9}$$

Denote $\mathbf{1} = (1, 1, \ldots, 1)^T$, then we have

$$0 \leq \mathbf{1}^T \Sigma \mathbf{1} = \sum_{i=1}^{m} \sum_{i'=1}^{m} \sigma_{ii'jk} = \sum_{i=1}^{m} \sigma_{ijk}^2 + \sum_{i \neq i'}^{m} \sigma_{ii'jk} \leq m\sigma_0^2 + \sum_{i \neq i'}^{m} \sigma_{ii'jk}, \tag{10}$$

Hence

$$0 \leq \frac{\mathbf{1}^T \Sigma \mathbf{1}}{m^2} \leq \frac{\sigma_0^2}{m} + \frac{\sum_{i \neq i'}^{m} \sigma_{ii'jk}}{m^2} = o(1). \tag{11}$$

Thus, for each $k = 1, 2, \ldots, n$, and for each taxon $i = 1, 2, \ldots, m$, according to Assumption 0.3, we have:

$$\frac{1}{m} \sum_{i=1}^{m} (y_{ijk} - (d_{jk} + \mu_{ij})) \to_p 0 \ \text{as} \ m \to \infty. \tag{12}$$

Thus,

$$\hat{d}_{jk} = \overline{y}_{\cdot jk} - \overline{y}_{\cdot j \cdot} \to_p (d_{jk} + \overline{\mu}_{\cdot j}) - (\overline{d}_{\cdot j} + \overline{\mu}_{\cdot j}) = d_{jk} - \overline{d}_{\cdot j}, \text{as} \ m \to \infty. \tag{13}$$

Using (8) and (13), let

$$\hat{\sigma}_{ij}^2 = \frac{1}{n^2} \sum_{k=1}^{n} (y_{ijk} - \hat{d}_{jk} - \hat{\mu}_{ij})^2 \tag{14}$$

denote the mean residual sum of squares. Then under some mild regularity conditions[24], we have the following consistency result

$$n(\hat{\sigma}_{ij}^2 - \sigma_{ij}^2) \to_p 0, \text{as} \ m, n \to \infty. \tag{15}$$

Therefore, using $\hat{\sigma}_{ij}$ for $\sigma_{ij}$ in (8) and appealing to Slutsky's theorem, we have:

$$\frac{\hat{\mu}_{ij} - (\mu_{ij} + \overline{d}_{j \cdot})}{\hat{\sigma}_{ij}} \to_d N(0, 1), \text{as} \ m, n \to \infty. \tag{16}$$

Furthermore, based on Assumption 0.3, from (8) and (15) we obtain:

$$\hat{\sigma}_{ij} \to_p 0, \text{as} \ m, n \to \infty. \tag{17}$$

Consequently,

$$\hat{\mu}_{ij} \to_p \mu_{ij} + \overline{d}_{j \cdot}, \text{as} \ m, n \to \infty. \tag{18}$$

The above observations regarding the convergence of various statistics play a critical role in the following. Since the sampling fraction is constant for all taxa within the subject, we attempt to pool information across taxa when estimating $\delta$. We model the taxa using the following Gaussian mixtures model. For the $i$th taxon, $i = 1, 2, \ldots, m$, let $\Delta_i = \hat{\mu}_{i1} - \hat{\mu}_{i2}$. Let $C_0$ denote the set of taxa that are not differentially abundant between the two groups, i.e., $C_0 = \{i \in (1, 2, \ldots, m): \mu_{i1} = \mu_{i2}\}$, $C_1$ denote the set of taxa whose mean abundance in group 1 is less than that of group 2, i.e., $C_1 = \{i \in (1, 2, \ldots, m): \mu_{i1} < \mu_{i2}\}$, and let $C_2$ denote the set of taxa whose mean abundance in group 1 is greater than that of group 2, i.e., $C_2 = \{i \in (1, 2, \ldots, m): \mu_{i1} > \mu_{i2}\}$, Let $\pi_r$ denote the probability that a taxon belongs to set $C_r$, $r = 0, 1, 2$. For simplicity of estimation of parameters, similar to GEE, we shall assume that $\Delta_i$, $i = 1, 2, \ldots, m$ are independently distributed. Thus, we ignore the underlying correlation structure when estimating $\delta$. This is similar to what is often done in other omics studies. Thus, we model the distribution of $\Delta_i$ by Gaussian

mixture as follows:

$$f(\Delta_i) = \pi_0 \phi\left(\frac{\Delta_i - \delta}{\nu_{i0}}\right) + \pi_1 \phi\left(\frac{\Delta_i - (\delta + l_1)}{\nu_{i1}}\right) + \pi_2 \phi\left(\frac{\Delta_i - (\delta + l_2)}{\nu_{i2}}\right), \tag{19}$$

where

(1) $\phi$ is the normal density function,
(2) $\delta + l_1$ and $\delta + l_2$ are means for $\Delta_i | C_1$, and $\Delta_i | C_2$, respectively. $l_1 < 0$, $l_2 > 0$,
(3) $\nu_{i0}$, $\nu_{i1}$, and $\nu_{i2}$ are variances of $\Delta_i | C_0$, $\Delta_i | C_1$, and $\Delta_i | C_2$, respectively.

For computational simplicity, we assume that $\nu_{i1} > \nu_{i0}$, $\nu_{i2} > \nu_{i0}$. Thus, without loss of generality for $\kappa_1$, $\kappa_2 > 0$, let $\nu_{i1} = \nu_{i0} + \kappa_1$ and $\nu_{i2} = \nu_{i0} + \kappa_2$. While this assumption is not a requirement for our method, it is reasonable to assume that variability among differentially abundant taxa is larger than that among the null taxa. By making this assumption, we speed-up the computation time.

Assuming samples are independent between experimental groups, we begin by first estimating $\nu_{i0}^2 = Var(\hat{\mu}_{i1} - \hat{\mu}_{i2}) = Var(\hat{\mu}_{i1}) + Var(\hat{\mu}_{i2})$. Using the estimator stated in (14), we estimate $\nu_{i0}^2$ consistently as follows:

$$\hat{\nu}_{i0}^2 = \sum_{j=1}^{2} \hat{\sigma}_{ij}^2 = \sum_{j=1}^{2} \frac{1}{n^2} \sum_{k=1}^{n} (y_{ijk} - \hat{d}_{jk} - \hat{\mu}_{ij})^2. \tag{20}$$

In all future calculations, we plug in $\hat{\nu}_{i0}^2$ for $\nu_{i0}^2$. This is similar in spirit to many statistical procedures involving nuisance parameters. The following lemma is useful in the sequel.

## Lemma 0.1.

$\frac{\partial}{\partial \theta} \log f(x) = E_{f(z|x)}[\frac{\partial}{\partial \theta} \log f(z) + \frac{\partial}{\partial \theta} \log f(x|z)]$.[25]

Let $\Theta = (\delta, \pi_1, \pi_2, \pi_3, l_1, l_2, \kappa_1, \kappa_2)^T$ denote the set of unknown parameters, then for each taxon the log-likelihood can be reformulated using Lemma 0.1, as follows:

$$\Theta \leftarrow \arg\max_\Theta \sum_{i=1}^{m} \sum_{r=0}^{2} p_{r,i}[\log \Pr(i \in C_r) + \log f(\Delta_i | i \in C_r)]. \tag{21}$$

Then the E–M algorithm is described as follows:

- E-step: Compute conditional probabilities of the latent variable. Define $p_{r,i} = \Pr(i \in C_r | \Delta_i) = \frac{\pi_r \phi\left(\frac{\Delta_i - (\delta + l_r)}{\nu_{ir}}\right)}{\sum_r \pi_r \phi\left(\frac{\Delta_i - (\delta + l_r)}{\nu_{ir}}\right)}$, $r = 0, 1, 2$; $i = 1, \ldots, m$, which are conditional probabilities representing the probability that an observed value follows each distribution. Note that $l_0 = 0$.

- M-step: Maximize the likelihood function with respect to the parameters, given the conditional probabilities.

We shall denote the resulting estimator of $\delta$ by $\hat{\delta}_{EM}$.

Next we estimate $Var(\hat{\delta}_{EM})$. Since the likelihood function is not a regular likelihood and hence it is not feasible to derive the Fisher information. Consequently, we take a simpler and a pragmatic approach to derive an approximate estimator of $Var(\hat{\delta}_{EM})$ using $Var(\hat{\delta}_{WLS})$, which is defined below. Extensive simulation studies suggest that $\hat{\delta}_{EM}$ and $\hat{\delta}_{WLS}$ are highly correlated (Supplementary Fig. 9) and it appears to be reasonable to approximate $Var(\hat{\delta}_{EM})$ by $Var(\hat{\delta}_{WLS})$.

Let $\{C_r\} = m_r$, $r = 0, 1, 2$, then

$$\hat{\delta}_{WLS} = \frac{\sum_{i \in C_0} \frac{\Delta_i}{\hat{\nu}_{i0}^2} + \sum_{i \in C_1} \frac{\Delta_i - \hat{l}_1}{\hat{\nu}_{i1}^2} + \sum_{i \in C_2} \frac{\Delta_i - \hat{l}_2}{\hat{\nu}_{i2}^2}}{\sum_{i \in C_0} \frac{1}{\hat{\nu}_{i0}^2} + \sum_{i \in C_1} \frac{1}{\hat{\nu}_{i1}^2} + \sum_{i \in C_2} \frac{1}{\hat{\nu}_{i2}^2}}$$
$$= \frac{\sum_{i \in C_0} \frac{\Delta_i}{\nu_{i0}^2} + \sum_{i \in C_1} \frac{\Delta_i - l_1}{\nu_{i1}^2} + \sum_{i \in C_2} \frac{\Delta_i - l_2}{\nu_{i2}^2}}{\sum_{i \in C_0} \frac{1}{\nu_{i0}^2} + \sum_{i \in C_1} \frac{1}{\nu_{i1}^2} + \sum_{i \in C_2} \frac{1}{\nu_{i2}^2}} + o_p(1). \tag{22}$$

The above expression is of the form

$$\frac{a_1^T x_1 + a_2^T x_2 + a_3^T x_3}{a_1^T \mathbf{1} + a_2^T \mathbf{1} + a_3^T \mathbf{1}} \equiv \frac{\alpha^T u}{\alpha^T \mathbf{1}}, \tag{23}$$

where

(1) $\mathbf{1} = (1, \ldots, 1)^T$,
(2) $a_r = (a_{r1}, a_{r2}, \ldots, a_{rm_r})^T := (\frac{1}{\nu_{ir}^2})^T$, $i \in C_r$, $r = 0, 1, 2$,
(3) $x_r = (x_{r1}, x_{r2}, \ldots, x_{rm_r})^T := (\Delta_i - l_i)^T$, $i \in C_r$, $r = 0, 1, 2$. Note that $l_0 = 0$,
(4) $\alpha = (\alpha_1, \alpha_2, \ldots, \alpha_m)^T \equiv (a_1^T, a_2^T, a_3^T)^T$,
(5) $u = (u_1, u_2, \ldots, u_m)^T \equiv (x_1^T, x_2^T, x_3^T)^T$.

For the simplicity of notation, we relabel $a$ and $x$ by $\alpha$ and $u$, respectively. Denote $Cov(x) = Cov(u)$ by $\Omega$, and let $\omega_{ii'}$ denotes the $(i, i')$ element of $\Omega$. As in Assumption 0.3, we make the following assumption

**Assumption 0.4.**

$$\frac{\sum_{i \neq i'}^{m} \omega_{ii'}}{m^2} = o(1). \tag{24}$$

Using the above expressions, we compute the variance as follows:

$$\text{Var}(\hat{\delta}_{\text{WLS}}) = \text{Var}\left(\frac{\alpha^T u}{\alpha^T \mathbf{1}}\right) = \frac{\sum_{i=1}^{m} \alpha_i^2 \omega_{ii}}{\left(\sum_{i=1}^{m} \alpha_i\right)^2} + \frac{\sum_{i \neq i'}^{m} \alpha_i \alpha_{i'} \omega_{ii'}}{\left(\sum_{i=1}^{m} \alpha_i\right)^2}. \tag{25}$$

Recall that (a) for $i \in C_0$, $\omega_{ii} = \text{Var}(\Delta_i) = \nu_{i0}^2 = O(n^{-1})$, (b) for $i \in C_1$, $\omega_{ii} = \text{Var}(\Delta_i) = \nu_{i1}^2 = \nu_{i0}^2 + \kappa_1 = O(1)$, and (c) for $i \in C_2$, $\omega_{ii} = \text{Var}(\Delta_i) = \nu_{i2}^2 = \nu_{i0}^2 + \kappa_2 = O(1)$. Note that $\alpha_i = \frac{1}{\text{Var}(\Delta_i)} = \frac{1}{\omega_{ii}}$, thus we have:

$$\text{Var}\left(\frac{\alpha^T u}{\alpha^T \mathbf{1}}\right) = \frac{\sum_{i=1}^{m} \alpha_i^2 \omega_{ii}}{\left(\sum_{i=1}^{m} \alpha_i\right)^2} + \frac{\sum_{i \neq i'}^{m} \alpha_i \alpha_{i'} \omega_{ii'}}{\left(\sum_{i=1}^{m} \alpha_i\right)^2} = \frac{1}{\sum_{i=1}^{m} \alpha_i} + \frac{\sum_{i \neq i'}^{m} \alpha_i \alpha_{i'} \omega_{ii'}}{\left(\sum_{i=1}^{m} \alpha_i\right)^2}. \tag{26}$$

Since $\nu_{i0}^2 = O(n^{-1})$, $\nu_{i1}^2 = O(1)$, and $\nu_{i2}^2 = O(1)$, consequently, $a_{1i} = O(n)$, $a_{2i} = a_{3i} = O(1)$, and

$$\begin{aligned}
\sum_{i=1}^{m} \alpha_i &= \mathbf{1}^T a_1 + \mathbf{1}^T a_2 + \mathbf{1}^T a_3 = \sum_{i \in C_0} O(n) + \sum_{i \in C_1} O(1) + \sum_{i \in C_2} O(1) \\
&= O(m_0 n) + O(m_1) + O(m_2) \\
&= O(m_0 n) \quad \text{if } m_0 n \geq \max\{m_1, m_2\}.
\end{aligned} \tag{27}$$

Using these facts and Assumption 0.4 in (26), we get

$$\begin{aligned}
\text{Var}\left(\frac{\alpha^T u}{\alpha^T \mathbf{1}}\right) &= O(m_0^{-1} n^{-1}) + \frac{\sum_{i \neq i'}^{m} \{n^{-1} m^{-1} \alpha_i\} \{n^{-1} m^{-1} \alpha_{i'}\} \omega_{ii'}}{n^{-2} m^{-2} \left(\sum_{i=1}^{m} \alpha_i\right)^2} \\
&= O(m_0^{-1} n^{-1}) + \frac{1}{m^2} \frac{\sum_{i \neq i'}^{m} \{n^{-1} \alpha_i\} \{n^{-1} \alpha_{i'}\} \omega_{ii'}}{\left(\sum_{i=1}^{m} n^{-1} m^{-1} \alpha_i\right)^2} \\
&= O(m_0^{-1} n^{-1}) + \frac{1}{m^2} \frac{O(1) o(m^2)}{O(1)} \\
&= O(m_0^{-1} n^{-1}).
\end{aligned} \tag{28}$$

Thus, under Assumption 0.4 regarding $\omega_{ii'}$, the contribution of the covariance terms in the above variance expression is negligible as long as $m$ is very large compared with $n$, which is usually the case. Hence

$$\text{Var}(\hat{\delta}_{\text{WLS}}) = \text{Var}\left(\frac{\alpha^T u}{\alpha^T \mathbf{1}}\right) = O(m_0^{-1} n^{-1}). \tag{29}$$

Furthermore, appealing to Cauchy–Schwartz inequality we get

$$\begin{aligned}
\text{Cov}(\hat{\mu}_{i1} - \hat{\mu}_{i2}, \hat{\delta}_{\text{WLS}}) &\leq \sqrt{\text{Var}(\hat{\mu}_{i1} - \hat{\mu}_{i2}) \text{Var}(\hat{\delta}_{\text{WLS}})} \\
&\leq O(n^{-1/2}) O(m_0^{-1/2} n^{-1/2}) = O(n^{-1} m_0^{-1/2}).
\end{aligned} \tag{30}$$

Hence, as long as $m_0$ is large, the contribution made by $\text{Var}(\hat{\delta}_{\text{WLS}})$ and $\text{Cov}(\hat{\mu}_{i1} - \hat{\mu}_{i2}, \hat{\delta}_{\text{WLS}})$ relative to $\text{Var}(\hat{\mu}_{i1} - \hat{\mu}_{i2})$ is negligible.

Neglect the covariance term in (26), let $\hat{C}_r$ denote the estimator of $C_r$, $r = 0, 1, 2$ from the E–M algorithm, define

$$\widehat{\text{Var}}(\hat{\delta}_{\text{WLS}}) = \frac{1}{\sum_{i \in \hat{C}_0} \frac{1}{\hat{\nu}_{i0}^2} + \sum_{i \in \hat{C}_1} \frac{1}{\hat{\nu}_{i1}^2} + \sum_{i \in \hat{C}_2} \frac{1}{\hat{\nu}_{i2}^2}}, \tag{31}$$

an estimator of $\text{Var}(\hat{\delta}_{\text{WLS}})$ under the Assumption 0.4. Then

$$\widehat{\text{Var}}(\hat{\delta}_{\text{WLS}}) \to_p \frac{1}{\sum_{i=1}^{m} \alpha_i} = \frac{1}{\sum_{i \in C_0} \frac{1}{\nu_{i0}^2} + \sum_{i \in C_1} \frac{1}{\nu_{i1}^2} + \sum_{i \in C_2} \frac{1}{\nu_{i2}^2}}, \text{ as } m, n \to \infty. \tag{32}$$

We performed extensive simulations to evaluate the bias and variance of $\hat{\delta}_{\text{EM}}$ and that of $\hat{\delta}_{\text{WLS}}$. The scatter plot (Supplementary Fig. 9) of $\hat{\delta}_{\text{EM}}$ and $\hat{\delta}_{\text{WLS}}$ are almost perfectly linear with correlation coefficient nearly 1 in all cases. This suggests that $\hat{\delta}_{\text{WLS}}$ is a very good approximation for $\hat{\delta}_{\text{EM}}$. The variance of $\hat{\delta}_{\text{EM}}$ as well as that of $\hat{\delta}_{\text{WLS}}$ are roughly of the order $n^{-1} m_0^{-1}$, as we expected. In addition, they are approximately unbiased (Supplementary Table 6).

**Hypothesis testing for two-group comparison.** For taxon $i$, we test the following hypothesis

$$\begin{aligned}
H_0 &: \mu_{i1} = \mu_{i2} \\
H_1 &: \mu_{i1} \neq \mu_{i2}
\end{aligned}$$

using the following test statistic which is approximately centered at zero under the null hypothesis:

$$W_i = \frac{\hat{\mu}_{i1} - \hat{\mu}_{i2} - \hat{\delta}_{\text{EM}}}{\sqrt{\hat{\sigma}_{i1}^2 + \hat{\sigma}_{i2}^2}}. \tag{33}$$

From Slutsky's theorem, we have:

$$W_i \to_d N(0, 1), \text{ as } m, n \to \infty. \tag{34}$$

If the sample size is not very large and/or the number of non-null taxa are very large, then we modify the above test statistic as follows:

$$W_i^* = \frac{\hat{\mu}_{i1} - \hat{\mu}_{i2} - \hat{\delta}_{\text{WLS}}}{\sqrt{\hat{\sigma}_{i1}^2 + \hat{\sigma}_{i2}^2 + \widehat{\text{Var}}(\hat{\delta}_{\text{WLS}}) + 2\sqrt{(\hat{\sigma}_{i1}^2 + \hat{\sigma}_{i2}^2)\widehat{\text{Var}}(\hat{\delta}_{\text{WLS}})}}}. \tag{35}$$

To control the FDR due to multiple comparisons, we recommend applying the Holm–Bonferroni method[26] or Bonferroni[27,28] correction rather than the Benjamini–Hochberg (BH) procedure[29] to adjust the raw $p$ values as research has showed that it is more appropriate to control the FDR when $p$ values were not accurate[30], and the BH procedure controls the FDR provided you have either independence or some special correlation structures such as perhaps positive regression dependence among taxa[29,31]. In our simulation studies, since the absolute abundances for each taxon are generated independently, we compared the ANCOM-BC results adjusted either by Bonferroni correction (Fig. 4) or BH procedure (Supplementary Fig. 10), it is clearly that the FDR control by Bonferroni correction is more conservative while implementing BH procedure results in FDR around the nominal level (5%). Obviously, ANCOM-BC has larger power when using BH procedure.

**Hypothesis testing for multigroup comparison.** In some applications, for a given taxon, researchers are interested in drawing inferences regarding DA in more than two ecosystems. For example, for a given taxon, researchers may want to test whether there exists at least one experimental group that is significantly different from others, i.e., to test

$$\begin{aligned}
H_{0,i} &: \cap_{j \neq j' \in \{1, \dots, g\}} \mu_{ij} = \mu_{ij'} \\
H_{1,i} &: \cup_{j \neq j' \in \{1, \dots, g\}} \mu_{ij} \neq \mu_{ij'}.
\end{aligned}$$

Similar to the two-group comparison, after getting the initial estimates of $\hat{\mu}_{ij}$ and $\hat{d}_{jk}$, setting the reference group $r$ (e.g., $r = 1$), and obtaining the estimator of the bias term $\hat{\delta}_{rj}$ through E–M algorithm, the final estimator of mean absolute abundance of the ecosystem (in log scale) are obtained by transforming $\hat{\mu}_{ij}$ of (6) into:

$$\hat{\mu}_{ij}^* := \begin{cases} \hat{\mu}_{ir}, & j = r \\ \hat{\mu}_{ij} + \hat{\delta}_{rj}, & j \neq r \in 1, \dots, g \end{cases}. \tag{36}$$

Thus, based on (18) and the E–M estimator of $\delta_{rj}$, as $m, \min(n_j, n_{j'}) \to \infty$

$$\hat{\mu}_{ij}^* - \hat{\mu}_{ij'}^* \to_p \begin{cases} 0 & \text{if taxon } i \text{ is not differentially abundant between group } j \text{ and } j', \\ \mu_{ij} - \mu_{ij'} & \text{otherwise}. \end{cases} \tag{37}$$

Similarly, the estimator of the sampling fraction is obtained by transforming $\hat{d}_{jk}$ of (6) into

$$\hat{d}_{jk}^* := \begin{cases} \hat{d}_{rk}, & j = r \\ \hat{d}_{jk} - \hat{\delta}_{rj}, & j \neq r \in 1, \dots, g \end{cases}. \tag{38}$$

As by (13) and the E–M estimator of $\delta_{rj}$

$$\hat{d}_{jk}^* \to_p d_{jk} - \overline{d}_r, \text{ as } m, \min(n_j, n_{j'}) \to \infty, \tag{39}$$

which indicates that we are only able to estimate sampling fractions up to an additive constant $(\overline{d}_r)$.

Define the test statistic for pairwise comparison as:

$$W_{i,jj'} = \frac{\hat{\mu}_{ij}^* - \hat{\mu}_{ij'}^*}{\sqrt{\hat{\sigma}_{ij}^2 + \hat{\sigma}_{ij'}^2}}, \quad i = 1, \dots, m, j \neq j' \in \{1, \dots, g\}. \tag{40}$$

For computational simplicity, inspired by the William's type of test[32–35], we reformulate the global test with the following test statistic:

$$W_i = \max_{j \neq j' \in \{1, \dots, g\}} |W_{i,jj'}|, \quad i = 1, \dots, m. \tag{41}$$

Under null, $W_{i,jj'} \to_d N(0, 1)$, thus we can construct the null distribution of $W_i$ by simulations, i.e., for each specific taxon $i$,

(a) Generate $W_{i,jj'}^{(b)} \sim N(0, 1), j \neq j' \in \{1, \dots, g\}, b = 1, \dots, B$.
(b) Compute $W_i^{(b)} = \max_{j \neq j' \in \{1, \dots, g\}} W_{i,jj'}^{(b)}$.
(c) Repeat above steps $B$ times (e.g., $B = 1000$), we then get the null distribution of $W_i$ by $(W_i^{(1)}, \dots, W_i^{(B)})^T$.

Finally, $p$ value is calculated as

$$p_i = \frac{1}{B} \sum_{b=1}^{B} I(W_i^{(b)} > W_i), \quad i = 1, \dots, m \tag{42}$$

and the Bonferroni correction is applied to control the FDR.

**Reporting summary.** Further information on research design is available in the Nature Research Reporting Summary linked to this article.

## Data availability

DNA sequences from the global gut microbiota study[9] can be found in MG-RAST https://www.mg-rast.org/index.html server under search string "mgp401" for Illumina V4-16S rRNA.

## Code availability

All analyses can be found under https://github.com/FrederickHuangLin/ANCOM-BC.

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

## Acknowledgements

This research was funded by the Department of Biostatistics, University of Pittsburgh, Pittsburgh, PA 15261, USA.

## Author contributions

Both authors contributed equally to the theory and methodology described in this paper. All numerical works and computations were conducted by H.L. who developed ANCOM-BC pipeline in R that is freely and publicly available. Please contact H.L. for software requests.

## Competing interests

The authors declare no competing interests.
