## [Peer Review File · Nature Communications]

Reviewers' Comments:

Reviewer #1:

Remarks to the Author:

NCOMMS-19-25269:
Analysis of Compositions of Microbiomes with
Bias Correction

Referee report for the authors

August 28, 2019

General assessment. I agree with the importance of the *sampling fraction* if one is interested in testing for differential abundance (DA) at the absolute level. The new method seem to perform well in the simulation study, although I have some thoughts about the interpretation and setup of the study (see further). Given the good results of the method, I would expect that there must be a good theoretical basis. However, I have many comments on the derivation of the method (see further).

Comments

1. One major novelty of the method is the bias correction. It is also argued in the paper that other tests for DA suffer from an uncontrolled FDR which gets worse with increasing sample sizes, and that this is caused by this bias (e.g. last sentences in the section “Differential abundance analysis”.) It would be helpful to give the reader here already some understanding of this bias.
2. I like figure 2, but from the text and the figure, it is still not completely clear to me what are the exact definitions of ecosystem, sample and specimen. These terms seem crucial to fully understand the paper. It would also be helpful to use the same terminology and reference to the figure, when defining the notations in Table 1.
3. I think that many people working in the field, are used to terms as “relative” and “absolute” abundance. Could the relationship between your terminology/notation be better clarified?
4. Related to the previous two comments: in table 1 you define A_{ijk} as the “expected” abundance, whereas I think it is a random variable (it is θ_{ij} which is the expected value of A_{ijk}). Do I get it right that A_{ijk}

is the absolute abundance (in a unit volume in sample k) and θ_{ij} is its expectation?

5. In table 1 you define $c_{jk} = \frac{O_{.jk}}{A_{.jk}}$, but Assumption 0.1 implies $c_{jk} = \frac{E\{O_{ijk}|A_{ijk}\}}{A_{ijk}}$. Which of the two is correct? Is the former an estimator of the latter? The development of the method seems to rely on Assumption 0.1 (this gives part of the statistical model), but in Equation (9) you explicitly use $c_{jk} = \frac{O_{.jk}}{A_{.jk}}$ (but with the index j dropped).
6. Equation (7): you cannot use the index k in the LHS of the equation, and as a running index in the summation at the LHS of the equation.
7. Lemma 0.3, condition (a). Should this hold for all $k = 1, 2, \dots$?
8. Equation (8): here you give a convergence result for when $m \rightarrow \infty$. The result of this convergence, which is given behind the arrow \rightarrow_p may thus no longer depend on m (because m is on its way to infinity). A correct statement is

$$\frac{1}{m} \sum_{i=1}^m (A_{ik} - \theta_i) \rightarrow_p 0.$$

This says that for large m , $\frac{1}{m} \sum_{i=1}^m A_{ik}$ can be approximated by $\frac{1}{m} \sum_{i=1}^m \theta_i = \bar{\theta}$. In particular, $\frac{1}{m} \sum_{i=1}^m A_{ik} = \bar{\theta} + o_p(1)$.

9. In Example 0.5 you give a decaying covariance, and starting from “Let $S_k = \dots$ you actually give part of the proof of Lemma 0.3. This is not necessary. If the conditions of the lemma are satisfied, you can simply refer to the lemma.
10. Equation (9): you use $c_{jk} = \frac{O_{.jk}}{A_{.jk}}$, but this not the same as Assumption 0.1. Upon using $E\{O_{ijk}\} = c_k \theta_i$ and $\frac{1}{m} \sum_{i=1}^m O_{ik} = \frac{1}{m} \sum_{i=1}^m (c_k \theta_i) + o_p(1)$, you can easily show

$$\frac{O_{.k}}{\frac{1}{n} \sum_{l=1}^n O_{.l}} \rightarrow_p \frac{c_k}{\frac{1}{n} \sum_{l=1}^n c_l} \quad \text{as } m \rightarrow \infty.$$

11. Equation (10). This result applies to the initial estimate $d_k^{(0)}$, but it still doesn't show that C is also the bias for the final estimator of d_k .
12. Equation (11) is important, because it contains the bias term C , which you will eliminate later on. However, Equation (11) falls out of the sky. Can you prove that C is also the bias of the estimator of μ_i ? A minor remark about Equation (11): it should be stated as an asymptotic result, i.e.

$$\frac{\hat{\mu}_i - (\mu_i + C)}{\sigma_i} \rightarrow_d N(0, 1) \quad \text{as } n \rightarrow \infty.$$

13. There is no need for Equation (15); you may immediately state the consistency of $\hat{\mu}_i$ for $\mu_i + C$ (Eq 16). The big question for me, however, is still where this bias C comes from. This is very important for what follows.
14. The description of the Gaussian mixture is mathematically very imprecise. For example, in Eq (19) you have clearly the density function of η_i as $f(\eta_i | Z_i = 1)$, but you state that this is equal to $\phi(l_1, v_1^2)$, which is no longer a function of η_i ! A correct statement is

$$f(\eta_i | Z_i = 1) = \phi\left(\frac{\eta_i - l_1}{v_1}\right),$$

with $\phi(\cdot)$ the density function of the standard normal distribution.

15. Equation (20) is again not correctly written. Apart from the ϕ -notation of my previous remark, it contains the term π_2 , which is not a function of η_i . I guess this π needs to be multiplied with a point probability at $\eta_i = 0$?
16. Can you clearly state all assumptions underlying the Gaussian mixture methodology for bias correction?
17. Lemma 0.6. No need to give a proof. This is a well known result. You may e.g. refer to:
McLachlan, G., & Krishnan, T. (2007). The EM algorithm and extensions (Vol. 382). John Wiley & Sons.
18. Equation (27): I am not convinced that this is correct. You use the consistency of $\hat{\delta}$, i.e. $\hat{\delta} \rightarrow_p \delta$ and Slutsky, but this is not sufficient. I think the reason why you did not see it, is that your notation for asymptotic statistics is not conventional as it does not explicitly shows the dependence on the sizes m , n_1 and n_2 . I give a very simple examen: the sample mean of n i.i.d. observations X_i with mean $\mu + \delta$ and variance σ^2 . We all know that the variance of the sample mean equals σ^2/n . Then we would write

$$\sqrt{n} \frac{\hat{\mu} - (\mu + \delta)}{\sigma} \rightarrow_p N(0, 1) \quad \text{as } n \rightarrow \infty.$$

Note that in this example $E\{\hat{\mu}\} = \mu + \delta$ and $\hat{\mu}$ is thus a biased estimator of μ .

Now suppose we have a consistent estimator of δ , just like you do. Thus $\hat{\delta}$ converges in probability to δ , but for getting the asymptotic distribution (convergence in distribution), we need to know the properties of $\sqrt{n}(\hat{\delta} - \delta)$. Often the convergence of estimators is of rate \sqrt{n} , which will result in $\sqrt{n}(\hat{\delta} - \delta)$ having a limit distribution (no longer convergence to a constant). Correcting the initial estimator $\hat{\mu}$, as you do, would give

$$\sqrt{n} \frac{\hat{\mu} + \hat{\delta} - (\mu + \delta)}{\sigma} = \sqrt{n} \frac{\hat{\mu} - \mu}{\sigma} + \sqrt{n} \frac{\hat{\delta} - \delta}{\sigma},$$

in which the first term has a $N(0, 1)$ limiting distribution, but also the second term has a limiting distribution (because of the factor \sqrt{n} and the rate of convergence of a regular estimator).

In conclusion, I think that your result is incorrect. In particular, I expect the variance of W_i to be wrong.

19. On p. 9 and p. 10 you write that you apply the Bonferroni method for controlling the FDR. Bonferroni is used for controlling the FWER. Perhaps you wanted to write: Benjamini and Hochberg?
20. You evaluate normalisation methods by means of the centered deviance, which you define in Suppl. Inf. Although I do not entirely disagree, I wonder whether this does not favour your own method, because you do apply an additive bias correction.
21. Suppl. Inf. Section 2: I did not read all of this section, because it focus on the consequences of the bias, and to me the bias is still not entirely clear (see earlier).
As a side note: there are again some notational issues in the statement of Lemma 2.1. For example, the convergence in distribution of $T_{\hat{\beta}_i}$ cannot have $\text{SE}(\hat{\beta}_i)$ at the RHS of the arrow (because the SE depends on n , which is on its way to infinity). Also note that you cannot write $\text{SE}(\hat{\beta}_i) = O_p(n^{-1/2})$, because the SE is not stochastic. You may write $O(n^{-1/2})$.
22. Simulation study (Suppl. Inf. Section 3), point (h)(ii): are the effect sizes relative? It may be helpful to describe some of the parameters in the simulation study by the notation you used earlier, e.g. $\mu_2 - \mu_1$.
23. Simulation study (Suppl. Inf. Section 3): you simulate data from a Gamma-Poisson distribution (Negative Binomial). So I think that you favour EdgeR and DESeq2. It would be nice to also simulate more realistic data, or assess the type I error control by starting from a real dataset and repeatedly permute the samples over two mock treatment groups (note that this procedure does not allow you to calculate the FDR as all taxa are not DA).
24. Simulation study (Suppl. Inf. Section 3): Can you provide a clear description of how you compute the FDR? I particularly wonder what you consider as a false positive. Is it when $\theta_{i2} \neq \theta_{i1}$. The θ -parameters refer to the expectation of the A_{ijk} , which I interpret as absolute abundances. In this case you will favour your method relative to e.g. EdgeR and DESeq2, because it is well known that these latter methods actually test hypotheses related to relative abundances. I am not saying that you may not define FDR in terms of the absolute abundances (I think this is what you are deliberately targeting), but you should at least clearly describe it, and include it in the discussion.

25. My last comment may perhaps result in a new, but similar method. I think that at some places you make an error related to expectations of logarithmically transformed random variables, particularly when going from your Assumptions 0.1 and 0.2 to the statistical model of section 0.4. Your assumptions imply

$$\log E\{O_{ijk}\} = \log \theta_{ij} + \log c_{jk} = \mu_{ij} + d_{jk}, \quad (1)$$

but you work with the statistical model (Eq (4) and (5))

$$E\{\log(O_{ijk})\} = \mu_{ij} + d_{jk}.$$

However, $\log E\{O_{ijk}\} \neq E\{\log(O_{ijk})\}$.

I therefore wonder whether part of your bias may come from this inequality?!

Note that Model (1) is a proper semiparametric statistical model, and its parameters can be estimated. And, just thinking out loud, I also wonder whether the efficiency of estimating d_{jk} could be improved by accounting for the library size . . .

Reviewer #2:

Remarks to the Author:

The manuscript "Analysis of composition of microbiomes with bias correction" describes a new compositional data-aware differential abundance method that uses bias correction to account for sampling efficiency differences. Current compositional data-aware methods are difficult to interpret and cannot be used as drop-in replacements for standard data analysis methods. However, 16S rRNA marker-gene sequence data is inherently compositional and should be treated as such. The two primary limitations of current compositional data-aware methods are the need for reference frames or reference taxa and lack of standard errors, uncertainties, or p-values. ANCOM-BC, the method described in this manuscript, aims to address these limitations and therefore is of great interest to the scientific community. ANCOM-BC additionally aims to correct for differences in sampling efficiencies, a well-known but rarely accounted for bias in metagenomics. The mathematical foundations for the ANCOM-BC method is well defined however the method evaluations using simulated and experimental data are lacking. Additional validation would strengthen the work. I don't have the relevant background to comment on the mathematical proofs or statistical validity of the standard error estimates or appropriateness of the method used to assign p-values. Though the results simulated data validation results are promising. The manuscript or Github repository documentation does not describe how to use the method or real data - limiting usability. The manuscript will help drive the field to improve usability compositional aware metagenomic data analysis methods. I commend the author's on the manuscript reproducibility, easily reproduced all of the results figures in the main manuscript and supplemental material. However, the github repository and software documentation is lacking in general. I recommend additional validation along with increased documentation regarding method usability. Specific major and minor revisions are recommended below.

Major Changes

1. Software usability: As currently presented the ANCOM-BC method is not easily incorporated into a standard marker-gene data analysis workflow. Please add documentation to the github repository and manuscript for how to use ANCOM-BC, preferably starting from standard metagenomic data formats such as a biom file or R data structures such as phyloseq (10.18129/B9.bioc.phyloseq) or MRexperiment (10.18129/B9.bioc.metagenomeSeq). Not necessary for publication, but converting the ANCOM-BC functions into a Bioconductor package (<https://www.bioconductor.org/developers/>) or qiime2 plug-in (<https://library.qiime2.org/plugins/>) would significantly increase usability and value to the community. The two ANCOM-BC functions take a number of arguments, the arguments should be documented in the github repository and manuscript supplemental material.
2. The centered difference analysis is an interested and novel approach to comparing normalization methods a more intuitive description of the centered difference analysis potentially including a diagram would aid in interpreting the results.
3. Simulation benchmarking results: The simulations presented demonstrates the ability of the ANCOM-BC to control the false discovery rate without loss of power relative to other commonly used methods. As presented the simulated data description is hard to follow a diagram of simulated experimental design would make it easier to follow. Additional simulations to further demonstrate ANCOM-BC performance, maybe include more challenging experimental designs to identify ANCOM-BC limitations, what about smaller sample sizes, lower coverage, multi-sample comparisons. It is arguably more important to know when it is not appropriate to use a method than when it is appropriate. The simulation based validation could further be improved with a

better description of the experimental design along with additional simulations.

4. Real data benchmarking: The real data analysis is hard to interpret. A more quantitative analysis of the NMDS analysis in Figure 4A would further demonstrate the value of the ANCOM-BC normalization method. Figure 4B: unclear what you are trying to show here. I recommend trying to replicate some of the evaluations performed in McMurdie and Holmes 2014 (<https://doi.org/10.1371/journal.pcbi.1003531>).

5. The discussion section is lacking. Please provide additional text related to the relative strengths and weaknesses of ANCOM-BC method, how someone would use ANCOM-BC, as well as plans for future development if any.

Minor Changes

1. Include the github url in the abstract and methods sections. To prevent URL root and allow readers access to the version of the code presented in the manuscript consider using figshare or zenodo to obtain a DOI for the version of the code presented in the manuscript (<https://guides.github.com/activities/citable-code/>).

Results - Normalization

2. Please elaborate on the meaning of the bolded sentence "As the microbial load ..." on page 2. Specifically, why normalization methods are only able to estimate sampling fractions up to a constant. Also, comment on the practical implications of this limitation.

Differential abundance analyses

3. Include the acronym for zero-inflated Gaussian mixture model like in figure 3

4. Please include a statement as to why does the compositional nature of metagenomic data bias differential abundance analyses but RNAseq compositional nature not bias differential expression analyses?

Illustration of gut microbiota data

5. Consider revising the statement "we wonder is the results would be different ..." to provide additional information regarding suggested parameter changes or expected results. e.g. "...optimizing hyperparameters may improve results..."

6. Provide additional justification or reference for using cutoffs for age group comparisons.

Methods

7. Justify selecting 1k library size for sample filtering (hardcoded value in ANCOM-BC script).

Response to reviewers' comments

We thank the reviewers for their careful reading of our manuscript and making several important comments that improved the presentation of the paper substantially. Reviewer 1 identified several important notations related issues that led to confusion and lack of clarity. We have addressed each of them. We hope that paper is easy to read and clear. We sincerely thank the reviewers for their very insightful and deep comments that have led to a substantially improved presentation of the paper. Major highlights of this revision are as follows:

1. As suggested by reviewer 2, we have included more simulation studies to evaluate the performance of ANCOM-BC. In every situation we considered ANCOM-BC provided the best control of FDR. The existing methods failed to control the FDR.
2. Reviewer 1 identified a number of technical issues which were partly due to poor choice of notations, typos and not clearly explaining the various terms. We took advantage of all the helpful suggestions made by the reviewer and re-wrote the methods section so that it is more precise and clear. Supplementary text provides theoretical explanations for why some of the existing methods fail to normalize the data for differential sampling fractions across samples.
3. As can be seen from Fig 3 (and Supplementary Fig S1, S2, S8), ANCOM-BC consistently does a good job in normalizing the data and eliminating the bias due to differential sampling fractions across specimens. Thus, by accurately estimating the bias to differential sampling fraction, ANCOM-BC controls the false discovery rate very well (Fig 4, Supplementary Figures 3-5). Other methods are not consistent in this regard (Figure 3, Figure S1) and hence fail to control FDR (Fig 4, Supplementary Figures 3-5).

Again, we thank the reviewers for their excellent comments that brought the best out of our paper.

In the following we provide item by item responses to all the comments we received. In each case the reviewers' comments (In italics) are followed by our responses. For brevity of document, we provided the key sentences of the reviewers' comments rather than typing the entire comment made by each reviewer.

Reviewer 1:

1. *One major novelty of the method is the bias correction. It is also argued in the paper that other tests for DA suffer from an uncontrolled FDR which gets worse with increasing sample sizes, and that this is caused by this bias (e.g. last sentences in the section "Differential abundance analysis".) It would be helpful to give the reader here already some understanding of this bias.*

Response: We have provided detailed explanation of bias in paragraph 6 in the Introduction section of main text, paragraph right below equation (7) in Methods section of main text, as well as Section 1 "Inflated false positive rates of some standard methods" of the Supplementary Information.

2. *I like figure 2, but from the text and the figure, it is still not completely clear to me what are the exact definitions of ecosystem, sample and specimen. These terms seem crucial to fully understand the*

paper. It would also be helpful to use the same terminology and reference to the figure, when defining the notations in Table 1.

Response: We have clarified the meanings of these terms in the new Figure 2, as well as in paragraph 5 in the Introduction section of main text.

3. *I think that many people working in the field, are used to terms as “relative” and “absolute” abundance. Could the relationship between your terminology/notation be better clarified?*

Response: We have clarified the meanings of these terms in the new Figure 1 and in the legend of Figure 1. We have replaced “abundance” by “absolute abundance” throughout the manuscript.

4. *Related to the previous two comments: in table 1 you define A_{ijk} as the “expected” abundance, whereas I think it is a random variable (it is θ_{ij} which is the expected value of A_{ijk}). Do I get it right that A_{ijk} is the absolute abundance (in a unit volume in sample k) and θ_{ij} is its expectation?*

Response: Sorry for the confusion. We agree, the table was not very clear. We hope the revised Table 1 along with equations (1) and (2) in Methods section make it clear what we mean.

5. *In table 1 you define $c_{jk} = O_{.jk}/A_{.jk}$, but Assumption 0.1 implies $c_{jk} = E(O_{ijk}|A_{ijk})/A_{ijk}$. Which of the two is correct? Is the former an estimator of the latter?*

Response: Again, we agree with the reviewer’s comment and suitably modified the definitions in Table 1 as well as elsewhere as necessary. We confirm the definition of $c_{jk} = E(O_{ijk}|A_{ijk})/A_{ijk}$, while $O_{.jk}/A_{.jk}$ is its empirical value.

6. *Equation (7): you cannot use the index k in the LHS of the equation, and as a running index in the summation at the LHS of the equation.*

Response: Our estimation equations have been simplified substantially and hence equations (6), (7), (8), (9), (10) and (11) are not relevant anymore or modified accordingly.

7. *Lemma 0.3, condition (a). Should this hold for all $k = 1, 2, \dots$?*

Response: Thanks for pointing it out! Lemma 0.3 has been replaced by Assumption 0.3.

8. *Equation (8): here you give a convergence result for when $m \rightarrow \infty$. The result of this convergence, which is given behind the arrow p may thus no longer depend on m (because m is on its way to infinity). A correct statement is*

$$\frac{1}{m} \sum_{i=1}^m (A_{ik} - \theta_i) \rightarrow_p$$

Response: Our estimation equations have been simplified substantially and hence equations (6), (7), (8), (9), (10) and (11) are not relevant anymore or modified accordingly.

9. *In Example 0.5 you give a decaying covariance and starting from “Let $S_k = \dots$ you actually give part of the proof of Lemma 0.3. This is not necessary. If the conditions of the lemma are satisfied, you can simply refer to the lemma.*

Response: We agree with reviewer’s comment. Lemma 0.3 has been replaced by Assumption 0.3, and Example 0.5 is no longer relevant.

10. *Equation (9): you use $c_{jk} = O_{.jk}/A_{.jk}$, but this is not the same as Assumption 0.1.*

Response: Again, we have followed reviewer’s suggestion and confirmed the definition of $c_{jk} = E(O_{ijk}|A_{ijk})/A_{ijk}$. Equations (6), (7), (8), (9), (10) and (11) are not relevant anymore or modified accordingly

11. *Equation (10). This result applies to the initial estimate $d_k^{(0)}$, but it still doesn’t show that C is also the bias for the final estimator of d_k .*

Response: Our estimations equations were modified, and we have eliminated the notation of “C” and simplified the descriptions accordingly. Please see equations (13) and (18) in Methods section of main text.

12. *Equation (11) is important, because it contains the bias term C, which you will eliminate later on. However, Equation (11) falls out of the sky. Can you prove that C is also the bias of the estimator of μ_i ?*

Response: We thank the reviewer for this comment. Equation (11) is no longer relevant, please see equations (13) and (18) in Methods section for asymptotic properties of \hat{d}_{jk} and $\hat{\mu}_{ij}$.

13. *There is no need for Equation (15), you may immediately state the consistency of $\hat{\mu}_i$ for $\mu_i + C$ (Eq 16). The big question for me, however, is still where this bias C comes from. This is very important for what follows.*

Response: We agree with the reviewer and accordingly made the change. Again, the comment regarding C does not apply anymore since we have addressed this issue as noted above. Please see equations (13) and (18) in Methods section.

14. *The description of the Gaussian mixture is mathematically very imprecise.*

Response: Thanks again. We have fixed the notations suitably to make it precise. Please see equation (19) in Methods section.

15. *Equation (20) is again not correctly written. Apart from the φ -notation of my previous remark, it contains the term π_2 , which is not a function of η_i . I guess this π needs to be multiplied with a point probability at $\eta_i = 0$?*

Response: Thanks again. As noted above we made the notations precise. Please see equation (19) in Methods section.

16. *Can you clearly state all assumptions underlying the Gaussian mixture methodology for bias correction?*

Response: We have now stated the assumptions explicitly. Please see the paragraph right below equation (19).

17. *Lemma 0.6. No need to give a proof. This is a well-known result.*

Response: Thanks. We have edited it accordingly. Please see Lemma 0.1 in Methods section.

18. *Equation (27): I am not convinced that this is correct. You use the consistency of $\hat{\delta}$, i.e. $\hat{\delta} \rightarrow_p \delta$ and Slutsky, but this is not sufficient.*

Response: The reviewer is correct. We made the correction to the test statistic in view of the reviewer's comment. Please see equations (35) in Methods section. It is important to note that the variance term associated with the estimator of bias is $O(1/nm_0)$, where m_0 is the number of non-differentially abundant taxa. Thus, if the number of non-differentially abundant taxa is large then in that case the variance of the estimator of the bias will be practically zero.

19. *On p. 9 and p. 10 you write that you apply the Bonferroni method for controlling the FDR. Bonferroni is used for controlling the FWER. Perhaps you wanted to write: Benjamini and Hochberg?*

Response: Thanks for the suggestion! Note that Bonferroni method uses a larger threshold than the BH procedure, thus making Bonferroni method more conservative than BH procedure for controlling FDR. The reason we deliberately chose Bonferroni method for controlling FDR is because it is robust to any dependence structure among the taxa, whereas BH procedure controls FDR for special forms of dependence which may or may not hold for microbiome data. Thus, we may be losing some power for using Bonferroni threshold instead of the BH, but we can be assured that we do not have to make any additional assumptions regarding the underlying dependence among microbiome.

20. *You evaluate normalisation methods by means of the centered deviance, which you denote in Suppl. Inf. Although I do not entirely disagree, I wonder whether this does not favour your own method, because you do apply an additive bias correction.*

Response: The method of evaluating the normalization methods is not designed to help our method. We think our previous description was a bit confusing to the reviewer. We have rewritten that section in Section 2 of Supplementary text under the title "Residual analysis of normalization methods for differential sampling fractions." Essentially, our proposed approach is akin to performing residual analysis in linear regression. Thus, all methods are on equal footing.

21. *Suppl. Inf. Section 2: I did not read all of this section, because it focusses on the consequences of the bias, and to me the bias is still not entirely clear (see earlier). As a side note: there are again some notational issues in the statement of Lemma 2.1.*

Response: We clearly stated the bias in paragraph “Bias and variance of bias estimation under the null hypothesis” in Methods section of main text. We also changed the notation from $SE(\hat{\beta}_i)$ to $\overline{SE}(\hat{\beta}_i)$ stated in Lemma 1.1 of Supplementary Information.

22. *Simulation study (Suppl. Inf. Section 3), point (h)(ii): are the effect sizes relative? It may be helpful to describe some of the parameters in the simulation study by the notation you used earlier.*

Response: In the revision we made it explicit as noted by the reviewer. The effect size is applied to the absolute abundance of group 1, defined relative to group 2. Reduction in absolute abundance of group 1, relative to group 2 corresponds to $U(0.1, 1)$. Increase in absolute abundance of group 1 relative to group 2 corresponds to $U(1, 10)$. We also provided the simulation flowcharts (Supplementary Fig. 10-12) using the notations described in Table 1.

23. *Simulation study (Suppl. Inf. Section 3): you simulate data from a Gamma-Poisson distribution (Negative Binomial). So I think that you favour EdgeR and DESeq2. It would be nice to also simulate more realistic data, of assess the type I error control by starting from a real dataset and repeatedly permute the samples over two mock treatment groups (note that this procedure does not allow you to calculate the FDR as all taxa are not DA).*

Response: As per the reviewer’s recommendation we have considered other simulation set-ups. We also simulated data using the “Global pattern” data set (Supplementary Fig. 5). The results are similar to what we saw from Poisson-Gamma simulation except more inflated FDR for DESeq2 and edgeR as expected.

24. *Simulation study (Suppl. Inf. Section 3): Can you provide a clear description of how you compute the FDR? In this case you will favour your method relative to e.g. EdgeR and DESeq2, because it is well known that these latter methods actually test hypotheses related to relative abundances.*

Response: Please see the simulation flowcharts (Supplementary Fig. 10-12) for illustrations on how we compute the FDR in our simulation studies. The two RNA-seq based methods DESeq2 and edgeR, are used in the literature to test null hypothesis regarding taxa abundances (Weiss et al., 2017). While the general impression is that edgeR and DESeq2 test hypothesis regarding relative abundance, however both methods first normalize the data by scaling them empirically derived constants and then test hypothesis (Chen, Yunshun, et al, 2014). The scaled parameters are interpreted as absolute abundances. However, as shown in the first section of Supplementary Information, these normalization factors do not estimate the difference in sampling fractions unbiasedly. This lack of bias correction results in inflated FDR. This is true with other methods as well, except for ANCOM-BC.

25. *My last comment may perhaps result in a new, but similar method. I think that at some places you make an error related to expectations of logarithmically transformed random variables, particularly when going from your Assumptions 0:1 and 0:2 to the statistical model of section 0:4.*

Response: We thank the reviewer for drawing our attention to the Jensen's inequality. However, as we noted in the previous draft of the paper, we take the same approach as the classical Box-Cox transformations when performing ANOVA or linear regression. Log-transformation is routinely performed in all theoretical and applied statistics. Interestingly, despite performing the log transformation on the data, our inferences about absolute abundance appear to be fine in terms of FDR control. Please see the paragraph right below the equation (5) in Methods section of main text.

Reviewer 2:

We thank the reviewer for his positive comments. In the following we provide item by item responses to his suggested major and minor changes.

Major changes

1. *Software usability: As currently presented the ANCOM-BC method is not easily incorporated into a standard marker-gene data analysis workflow. Please add documentation to the GitHub repository and manuscript for how to use ANCOM-BC*

Response: We thank the reviewer for his useful suggestions. We have added a user manual and demo to the Github repository with all arguments and values being documents. We will try to transform ANCOM-BC function to a QIIME2 plug-in in the near future.

2. *The centered difference analysis is an interested and novel approach to comparing normalization methods a more intuitive description of the centered difference analysis potentially including a diagram would aid in interpreting the results.*

Response: We thank the reviewer for his comment. As per his suggestion as well as reviewer 1's suggestion, we have made suitable changes. Please see Section 2 of Supplementary Information for the definition of centered deviance, which we now refer to as residual. Supplementary Table 7 for a summary of sampling fraction estimator for each method, and Supplementary Fig. 10-12 for simulation flowcharts.

3. *Simulation benchmarking results: As presented the simulated data description is hard to follow a diagram of simulated experimental design would make it easier to follow. Additional simulations to further demonstrate ANCOM-BC performance, maybe include more challenging experimental designs to identify ANCOM-BC limitations, what about smaller sample sizes, lower coverage, multi-sample comparisons.*

Response: Again, we thank the reviewer for several useful suggestions. We have now prepared flowcharts (Supplementary Fig. 10-12) of the simulation experiments. In response to this reviewer's suggestion as well as the suggestion of reviewer 1, we considered additional simulation studies. In addition to Poisson-Gamma, we also conducted a simulation study based on the real "Global pattern" data (Supplementary Fig. 5). We also considered simulation settings when the sample size was $n = 5, 10$, which is very small (See Supplementary Fig. 6). We also considered the cases when the fraction of true positives is large (e.g. 50% and 75%) (See Supplementary Fig. 7). For small sample size such as $n=5$, ANOCM-BC fails to control FDR, however it recovers well at $n=10$. When the fraction of true positives is as high as 75%, none of the methods control the FDR, they all inflate the FDR by quite a bit. However, even in these situations, ANCOM-BC and ANCOM have FDR within one SE or barely over one SE.

4. *Real data benchmarking: The real data analysis is hard to interpret. A more quantitative analysis of the NMDS analysis in Figure 4A would further demonstrate the value of the ANCOM-BC normalization method. Figure 4B: unclear what you are trying to show here.*

Response: We agree with this reviewer's opinion that Fig. 4b does not convey too much extra information and thus it has been removed. We re-arranged figures in the main text and quantified Fig. 5 (previously Fig. 4a) using between group sum-of-squares (SSB). The larger SSB, the more segregated of different clusters. ANCOM-BC outperforms other methods by having the largest SSB.

5. *The discussion section is lacking. Please provide additional text related to the relative strengths and weaknesses of ANCOM-BC.*

Response: We have now included a detailed discussion section addressing the issues raised by the reviewer.

Minor changes

1. *Include the GitHub URL in the abstract and methods section.*

Response: We have now addressed this suggestion. We provided a hidden link of DOI in the abstract, and Github URL under "Code Availability" below Methods section.

2. *Please elaborate on the meaning of the bolded sentence "As the microbial load ..." on page 2. Specifically, why normalization methods are only able to estimate sampling fractions up to a constant. Also, comment on the practical implications of this limitation*

Response: We have now addressed this suggestion. Please see equation (39) in Methods section of main text and section 2 of Supplementary Information for the reason why normalization methods are only able to estimate sampling fractions up to a constant.

3. *Include the acronym for zero-inflated Gaussian mixture model like in figure 3.*

Response: Done

4. *Please include a statement as to why does the compositional nature of metagenomic data bias differential abundance analyses but RNA-Seq compositional nature not bias differential expression analyses?*

Response: Please see section 1&3 of Supplementary Information. We believe the compositional nature will affect the DE analysis of RNA-Seq data as well. As demonstrated in section 1 of Supplementary Information, scaling methods implemented in DESeq2 and edgeR are not able to estimate the difference of sampling fractions unbiasedly, hence it will have impact on DE analysis.

5. *Consider revising the statement "we wonder is the results would be different ..." to provide additional information regarding suggested parameter changes or expected results. e.g. "...optimizing hyperparameters may improve results..."*

Response: We removed the original Fig. 4b, and this statement is no longer relevant.

6. *Provide additional justification or reference for using cutoffs for age group comparisons.*

Response: Since the gut microbiota evolve with age (Lozupone et al., 2013), it is more reasonable to perform DA analyses by different age groups. While choosing cutoff values at 2 and 18 is more data-driven, as it makes the sample size of Malawi, Venezuela, and the US more balanced.

7. *Justify selecting 1k library size for sample filtering (hardcoded value in ANCOM-BC script).*

Response: Thanks for catching this! We have provided a new argument in the pre-processing function, so users are able to specify threshold for filtering small library sizes.

Reviewers' Comments:

Reviewer #1:

Remarks to the Author:

NCOMMS-19-25269: Revision 1

Analysis of Compositions of Microbiomes with Bias Correction

Referee report for the authors

March 8, 2020

I would like to thank the authors for their thorough revision of their manuscript. The formulations are now much clearer. Despite the improvements to the text and the good simulation results, I still have a few fundamental comments which boil down to the core of the method. Below I give a detailed description of the issue, but in summary, I think that your regression model should not give biased estimators.

Problem with model formulation

I start from your model (5) for the log-transformed observations, $y_{ijk} = \log O_{ijk}$. From this model you derive the least squares estimators in your equation (6). I think it is here the issue starts. Immediately below equation (6) you write

$$E\{\hat{\mu}_{ij}\} = E\{\bar{y}_{\cdot jk}\} = \mu_{ij} + \bar{d}_j.$$

and because $E\{\hat{\mu}_{ij}\} \neq \bar{d}_j$, you conclude that $\hat{\mu}_{ij}$ is a biased estimator.

The problem starts with your model formulation (5). This model requires restrictions to make the d and μ parameters estimable. Let's look at a very simple example in which $i = 1, 2$ and $k = 1, 2$ (I will ignore the group index j because this does not affect the results). In this simple setting, the model in matrix notation becomes

$$E\left\{\begin{pmatrix} y_{11} \\ y_{12} \\ y_{21} \\ y_{22} \end{pmatrix}\right\} = \begin{pmatrix} 1 & 0 & 1 & 0 \\ 1 & 0 & 0 & 1 \\ 0 & 1 & 1 & 0 \\ 0 & 1 & 0 & 1 \end{pmatrix} \begin{pmatrix} d_1 \\ d_2 \\ \mu_1 \\ \mu_2 \end{pmatrix}.$$

Let

$$\mathbf{X} = \begin{pmatrix} 1 & 0 & 1 & 0 \\ 1 & 0 & 0 & 1 \\ 0 & 1 & 1 & 0 \\ 0 & 1 & 0 & 1 \end{pmatrix}$$

denote the design matrix. At this stage you can also see the issue. In each row the sum of the elements equals 1, i.e. the matrix \mathbf{X} is not of full rank and hence there is no unique solution for the parameter estimators. These estimators (in matrix notation), according to the usual least-squares theory, can be estimated as (with $\mathbf{y}^t = (y_{11}, y_{12}, y_{21}, y_{22})$),

$$(\mathbf{X}^t \mathbf{X})^{-1} \mathbf{X}^t \mathbf{y},$$

in which the matrix $\mathbf{X}^t \mathbf{X}$ is not invertible because \mathbf{X} is not of full rank.

I don't know exactly why you did not have discovered this problem. Perhaps it is your equation (6), which are NOT exactly the estimating equations. I don't know what went wrong here; maybe you substituted d_{jk} along the way already with \hat{d}_{jk} ? The correct estimating equations are given by $(\mathbf{X}^t \mathbf{X})\boldsymbol{\beta} = \mathbf{X}^t \mathbf{y}$ (with $\boldsymbol{\beta}^t = (d_1, d_2, \mu_1, \mu_2)$).

The solution is simple. For example, you can impose the restrictions (again ignoring j)

$$\sum_j d_j = 0.$$

With this restriction the model becomes

$$\mathbb{E} \left\{ \begin{pmatrix} y_{11} \\ y_{12} \\ y_{21} \\ y_{22} \end{pmatrix} \right\} = \begin{pmatrix} 1 & 1 & 0 \\ 1 & 0 & 1 \\ -1 & 1 & 0 \\ -1 & 0 & 1 \end{pmatrix} \begin{pmatrix} d_1 \\ -d_1 \\ \mu_1 \\ \mu_2 \end{pmatrix}.$$

And now the design matrix

$$\mathbf{X} = \begin{pmatrix} 1 & 1 & 0 \\ 1 & 0 & 1 \\ -1 & 1 & 0 \\ -1 & 0 & 1 \end{pmatrix}$$

is of full rank and the least-squares estimator exists and is unbiased (classical theory applies).

Note that it may be weird to only imply the sum-zero restriction $\sum_j d_j = 0$ to the d -parameters. You can also apply a restriction to the μ -parameters, but then you need an intercept in the model.

What is your bias correction?

Based on my previous comment, I now don't know what your bias estimator is estimating. Perhaps it is a bias, because your estimating equations (6) are not correct. However, even if this is the case, then I think that your approach is making things more complicated than they are.

One more comment on the mixture model approach: this method relies heavily on the normality assumption of the Δ_i (distribution over the taxa). So one may wonder whether the estimates of δ are really bias estimates, or estimates that simply make the distributions of the Δ_i look more normal?

Why does your method perform well?

Despite my comments, the simulation results demonstrate a very good performance of your method. Maybe this is because at the end your method works (bias correction is correction for poor estimation of model parameters, and the idea of working with a *sampling fraction* is surely a very good idea).

On the other hand, I have a problem with the use of the Bonferroni correction. In my first review, I asked about this (my comment 19), because I thought that is was a typo. I thought you meant the Benjamini-Hochberg correction for FDR control. However, in your reply to my comment, you confirmed the use of Bonferroni. Looking at your simulation results I see that your test is conservative (true FDR much smaller than nominal FDR level), which is what I indeed expect when using Bonferroni for 500 simultaneous tests. Still I think you should not use Bonferroni if your aim is controlling the FDR. Now it raises the questions how your method will perform if an appropriate FDR control method is used (such as Benjamini-Hochberg). What if the use of BH would make your method no longer control the FDR? Thus, in the current version of the manuscript, there is no fair comparison between the new test and the others.

Log-transformation

Going back to my original comment 25, which relates to the use of the log-transformation. You replied that a log-transformation is common in statistical data analysis and you referred to the Box-Cox transformation. This is all true, but this does not mean that this is a correct procedure. Particularly in your setting, in which bias is the target. By modelling $E\{\log O_{ijk}\} = d_{jk} + \mu_{ij}$, whereas the parameters only make sense for $\log(E\{O_{ijk}\}) = d_{jk}^* + \mu_{ij}^*$ (I've added * to make the distinction), we now from the start that parameter estimators of μ_{ij} may be biased estimators for μ_{ij}^* .

If I did not see problems in the other parts of your method, I would probably not find the log-transformation problematic, but given my earlier concerns, I cannot assess to what extent this log-transformation is adding up to the problem.

Reviewers' Comments:

Reviewer #1:

Remarks to the Author:

Reviewer #2:

Remarks to the Author:

The authors have done a nice job addressing the mine and other other reviewer's comments. I appreciate the additional documentation, simulations, and methods clarifications. A few remaining issues below.

Major Change

=====

- Supplemental figure 8 presents results for the simulation with large sampling fraction variability. This is the one simulation where ANCOM-BC has similar performance to other methods, yet it is not referenced or discussed in the manuscript. The results of this simulation should be included in the main text as they will help users determine if ANCOM-BC is appropriate for their specific dataset.
- Information provided in the response to reviewers regarding failure of ANCOM-BC to control FDR for sample sizes less than 10 should be included in the main text as it will help reader understand ANCOM-BC's limitations.

Minor Change

=====

- Figure 3 - change y-axis title to residual instead of centered deviance.
- Reference flowchart supplemental figures in normalization methods section.

Suggested Change

=====

- The additional documentation will help users run the software and a qiime2 plug-in will greatly improve usability. However, I still recommend making the code available as a R bioconductor package utilizing standard bioconductor data structures for non-qiime2 users. As the software is currently documented it would take a bit of work for a user to generate appropriately formatted inputs from standard microbiome data formats, e.g. biom files for phyloseq class objects, and will likely significantly impact the software's adoption, regardless of the method's validity.

Response to reviewer 1's comments

1. Problem with model formulation and what is your bias correction?

We believe that this reviewer has misunderstood the model and the problem at hand. There is a fundamental distinction between our model and the traditional ANOVA model considered by the reviewer. Let us consider the following simplified set-up. Suppose there are two experimental groups, say control and treatment groups, and a single outcome variable (denoted by Y). Suppose we have n subjects randomly allocated to each group. Then the traditional one way ANOVA model (which yields t-test in this simple example) is of the form:

$$Y_{jk} = d + \mu_j + \epsilon_{jk}, j = 1, 2, k = 1, 2, \dots, n. \quad (1)$$

Then, $\bar{Y}_j = \frac{\sum_{k=1}^n Y_{jk}}{n}$, $E(\bar{Y}_j) = d + \mu_j$, and $E(\bar{Y}_1 - \bar{Y}_2) = \mu_1 - \mu_2$. Under the null hypothesis of no treatment effect, i.e. $H_0: \mu_1 = \mu_2$, we have $E(\bar{Y}_1 - \bar{Y}_2) = 0$. Thus, under the null hypothesis, the numerator of the standard t-test is centered at 0.

However, in the present situation of microbiome data, each subject potentially has a different sampling fraction. Consequently, the constant d in model (1) varies with subject. Thus, model (1) is modified as

$$Y_{jk} = d_{jk} + \mu_j + \epsilon_{jk}, j = 1, 2, k = 1, 2, \dots, n. \quad (2)$$

Then $\bar{Y}_j = \frac{\sum_{k=1}^n Y_{jk}}{n}$, $E(\bar{Y}_j) = \bar{d}_j + \mu_j$, and $E(\bar{Y}_1 - \bar{Y}_2) = (\bar{d}_1 - \bar{d}_2) + (\mu_1 - \mu_2)$. Under the null hypothesis no treatment effect, i.e. $H_0: \mu_1 = \mu_2$, we have $E(\bar{Y}_1 - \bar{Y}_2) = (\bar{d}_1 - \bar{d}_2)$, which is the bias we are referring to! Thus, under the null hypothesis, the numerator of the test statistic is no longer centered at 0 unless we estimate the bias $(\bar{d}_1 - \bar{d}_2)$ and eliminate it. Alternatively, one needs to assume that $\bar{d}_1 = \bar{d}_2$ for all random samples, which may not be true.

Thus, the problem we considered is NOT the standard problem. We did not complicate the problem for the sake of complicating it nor because we did not think about the issue raised by the reviewer. He simply missed our point. We start our solution like the standard least squares problem (see equation (6) in our paper), similar to reviewer's comment about constraints etc. Actually, the equation (6) in the Methods section already shows that $\sum_k \hat{d}_{jk} = 0$. However, to estimate the bias $(\bar{d}_1 - \bar{d}_2)$ we recognize that d_{jk} are subject specific and not taxa specific and hence we can borrow information across taxa to estimate the bias. We used mixture model to do this estimation.

Here is the second misunderstanding by the reviewer. He questions the normality assumption when constructing the mixture model. Note that the mixture model was on the sample mean differences. The normal approximation is perfectly reasonable as it is inspired by the central limit theorem on the means.

2. **Why does your method perform well?**

Our method performs well because we are able to estimate the bias very well and eliminate it from the test statistic. Thus under the null hypothesis, our test statistic appropriately centered at zero, unlike all other tests which are centered incorrectly because their normalization methods do not correct the inherent bias in these microbiome data. Once again, the reviewer is incorrect in challenging us for not using the Benjamini-Hochberg (BH) procedure for FDR control. Previous studies have shown that the Bonferroni correction is more appropriate to control the FDR when p-values were not exact (Lim et al., 2013). Note that our test statistic is asymptotically unbiased. Even if the underlying data are independently distributed, due to the compositionality introduced by the sampling fractions, the test statistics (and the resulting p-values) for various taxa are NOT necessarily independently distributed. The BH procedure controls the FDR provided you have either independence or some special correlation structures such as perhaps positive regression dependence. Since, we cannot be sure about dependence structure we used the Bonferroni procedure, which is appropriate for all correlation structures but might be conservative. We did not mind losing power as long as we can control the FDR. Indeed our method does a very good job of controlling the FDR while competing very well with other methods in terms of power. However, for the sake of reviewer's comment, we have included Supplementary Fig. 10 in the Supplementary Information where all methods, including ANCOM-BC, use the BH procedure. Again, we notice that the ANCOM-BC continues to perform very well with FDR around 5%, and achieving highest power among all methods! Despite this excellent performance by ANCOM-BC using the BH procedure, we still prefer the conservative Bonferroni method because it protects the FDR when there are complex dependencies among the microbes. In the Methods section, preceding the section "Hypothesis testing for multi-group comparison", we added a sentence about the performance of ANCOM-BC when BH procedure was used.

3. **Log-transformation**

We made further clarifications under equation (5). We agree with the reviewer that we could have used slightly different notation in the log-linear model. However, we did not feel it was necessary because we did not want to inundate the reader with too many notations.

Response to reviewer 2's comments

1. **Supplemental figure 8 presents results for the simulation with large sampling fraction variability. This is the one simulation where ANCOM-BC has similar performance to other methods, yet it is not referenced or discussed in the manuscript. The results of this simulation should be included in the main text as they will help users determine if ANCOM-BC is appropriate for their specific dataset.**

We thank the reviewer for his comment. The discussion of the limitations using ANCOM-BC is shown in the 3rd paragraph of the Discussion section. We reference Supplementary Fig. 7 there for the simulation results where a large sampling fraction variability is present. We are reluctant to add the Supplementary Fig. 7 into main text (but keep it in the Supplementary Information) to avoid duplication in style with Fig. 4.

2. **Information provided in the response to reviewers regarding failure of ANCOM-BC to control FDR for sample sizes less than 10 should be included in the main text as it will help reader understand ANCOM-BC's limitations.**

Again, we thank the reviewer for his suggestion. The discussion of the limitations using ANCOM-BC (including the case when the same sizes are less than 10) is shown in the 3rd paragraph of the Discussion section.

3. **Figure 3 - change y-axis title to residual instead of centered deviance.**

Done.

4. **Reference flowchart supplemental figures in normalization methods section.**

Done.

5. **The additional documentation will help users run the software and a qiime2 plug-in will greatly improve usability. However, I still recommend making the code available as a R Bioconductor package utilizing standard Bioconductor data structures for non-qiime2 users. As the software is currently documented it would take a bit of work for a user to generate appropriately formatted inputs from standard microbiome data formats, e.g. biom files for phyloseq class objects, and will likely significantly impact the software's adoption, regardless of the method's validity.**

We really appreciate the reviewer's suggestion. Yes, we are planning to make the algorithm available on R Bioconductor, as well as make it a qiime2 plug-in after mid-May. We are waiting for this paper to be accepted for publication. Stay tuned!